DOI: 10.1038/s41467-018-05825-x | OPEN

# Insights into degradation mechanism of N-end rule substrates by p62/SQSTM1 autophagy adapter

Do Hoon Kwon [1], Ok Hyun Park[1,2], Leehyeon Kim [1], Yang Ouk Jung [1], Yeonkyoung Park[1,2], Hyeongseop Jeong[3], Jaekyung Hyun[3], Yoon Ki Kim [1,2] & Hyun Kyu Song [1]

p62/SQSTM1 is the key autophagy adapter protein and the hub of multi-cellular signaling. It was recently reported that autophagy and N-end rule pathways are linked via p62. However, the exact recognition mode of degrading substrates and regulation of p62 in the autophagic pathway remain unknown. Here, we present the complex structures between the ZZ-domain of p62 and various type-1 and type-2 N-degrons. The binding mode employed in the interaction of the ZZ-domain with N-degrons differs from that employed by classic N-recognins. It was also determined that oligomerization via the PB1 domain can control functional affinity to the R-BiP substrate. Unexpectedly, we found that self-oligomerization and disassembly of p62 are pH-dependent. These findings broaden our understanding of the functional repertoire of the N-end rule pathway and provide an insight into the regulation of p62 during the autophagic pathway.

[1] Department of Life Sciences, Korea University, Seoul 02841, Korea. [2] Creative Research Initiatives, Center for Molecular Biology of Translation, Korea University, Seoul 02841, Korea. [3] Electron Microscopy Research Center, Korea Basic Science Institute, Chungcheongbuk-do 28119, Korea. Correspondence and requests for materials should be addressed to H.K.S. (email: hksong@korea.ac.kr)

Protein homeostasis plays a fundamental role in cellular physiology and is strictly regulated by two different types of catabolic pathways: the ubiquitin-proteasome system (UPS) and the autophagy-lysosome system (ALS)[1–4]. There is growing evidence to suggest that these two systems communicate with each other to coordinate cellular degradation processes[5–7]. Intriguingly, the autophagy adapter p62/SQSTM1/Sequestosome-1 was recently reported to recognize N-degrons, the N-end rule substrates of the well-characterized UPS, and where ultimately these substrates are delivered to ALS[8,9]. p62 is a key selective autophagy adapter that plays a role in the degradation of various cellular constituents such as misfolded proteins and their aggregates, malfunctioning organelles, and invading pathogens[10–13]. It has been known that p62 acts as a signaling hub residing in the late endosome and lysosome[14], and is involved in various pathways related to human diseases[15–19].

p62 consists of six well-defined structural elements including Phox and Bem1p (PB1), ZZ-type zinc finger (ZZ), TRAF6-binding (TB), LC3-interacting region (LIR), Keap1-interacting region (KIR), and ubiquitin-associated domain (UBA)[20] (Fig. 1a). The N-terminal PB1 domain is responsible for oligomerization of p62, which is critical for its function and localization, the TB domain binds to TRAF6 for modulating TNF-α signaling, LIR is utilized for LC3-binding, which is critical for autophagy, and KIR is employed for regulating the Keap1-Nrf2 pathway, which is linked to major oxidative stress responses[21]. A great deal of attention has been devoted to investigating the role of ubiquitin (Ub) in selective autophagy besides its participation in the proteasomal degradation system[22,23], and the C-terminal UBA domain of p62 is believed to play a role in this process[24]. Intriguingly, it was recently reported that the central ZZ-domain in p62 plays a critical role in the recognition of N-terminal arginylated BiP/GRP78 by the Arg-tRNA transferase ATE1 (see ref.[8]). Therefore, this domain is particularly important for redirecting N-end rule substrates to the autophagy pathway.

The N-end rule pathway comprises a set of Ub-mediated protein degradation processes which controls the in vivo half-life of proteins depending on their N-terminal residue[25–27]. In eukaryotes, the N-end rule pathway comprises three classes, Arg/N-end, Ac/N-end, and the very recently identified Pro/N-end rule pathway[28]. The Arg/N-end rule was the first characterized pathway and targets proteins with the following primary N-terminal residues: type-1 (Arg, Lys, and His; positively charged residues recognized by the UBR box) and type-2 (Phe, Tyr, Trp, Leu, and Ile; bulky hydrophobic residues recognized by the ClpS-homology domain) N-degrons. Furthermore, it is organized in hierarchical steps whereby tertiary destabilizing N-terminal Asn and Gln residues of N-end rule substrates are deamidated to secondary destabilizing Asp and Glu residues, and the Arg residue is subsequently attached to these destabilizing residues. A set of endoplasmic reticulum (ER)-associated proteins, such as BiP/GRP78, calreticulin and protein disulfide isomerase, undergoes post-translational modification involving cleavage of a signal sequence by specific proteases, thereby exposing negatively charged residues[8]. In particular, the ATE1 enzyme in the N-end rule pathway adds an Arg residue at the new N-terminus of BiP, a chaperone that binds to misfolded protein aggregates. The supramolecular complex between BiP chaperone molecules and protein aggregates is recognized by the ZZ-domain of p62 (see ref.[8]), and ultimately this multi-protein complex is delivered to autophagosomes and degraded by lysosomes[10].

p62 is an enigmatic molecule that participates in many different cellular processes pertaining to protein homeostasis[17,29]. A proposed overall structure of p62 assumed a long helical filament structure via the PB1 domain[30], and the other domains TB, LIR, KIR, and UBA have been relatively well studied[20]. However, the function of the ZZ-domain has just begun to be explored and the manner by which Arg/N-end rule substrates are recognized remains unknown. Furthermore, the interplay between the PB1 and ZZ-domains has yet to be extensively investigated.

Here, we present the high resolution structures of the ZZ-domain of p62 in complex with 8 different N-degrons including type-1 and type-2, and subsequently identify key determinants involved in the unusual recognition. Subsequent biochemical and biophysical studies with p62 and N-degrons demonstrated a critical role of oligomerization mediated by the PB1 domain. Furthermore, it was unexpectedly found that self-oligomerization and disassembly of p62 are essentially controlled by pH. These findings provide fundamental insights into the manner by which a variety of N-end rule substrates are recognized by the ZZ-domain in addition to the role played by p62 in the whole autophagy pathway.

## Results

**Structure of the ZZ-domain of p62.** The structure of the ZZ-domain of human p62 (residues 126–172) was determined by a single-wavelength anomalous dispersion method at the zinc absorption edge (Fig. 1b and Supplementary Table 1). The negatively charged patch is formed by three β-strands, one α-helix, and two zinc atoms (Fig. 1c). As with previously known ZZ-domain structures, the zinc-coordinating residues are strictly conserved (Fig. 1d) and are located in zig-zag order for the first zinc atom (Zn1) coordinated by four cysteine residues and the second zinc atom (Zn2) coordinated by two cysteine and two histidine residues (Fig. 1d). The N-terminal U-shaped loop of the ZZ-domain is maintained by Cys128 and Cys131 residues coordinating to a zinc atom. One side of the protein surface is covered by a highly negatively charged patch (Fig. 1b) formed by four key residues (Asp129, Asn132, Asp147, and Asp149), which is more narrow and shallow compared to previously determined structures of N-recognins (UBR box)[31,32]. These four residues are highly conserved among p62 proteins (Supplementary Fig. 1a), but not in other ZZ-domain proteins (Fig. 1d). Among these, Asn132 completely differs with other ZZ-domains[33,34], and even in frog and zebrafish p62 this residue is replaced with Gln and Asp, respectively (Supplementary Fig. 1a). Furthermore, although NBR1 is a similar type of autophagy receptor that contains the ZZ-domain, the ZZ-domain of NBR1 possesses divergent residues and thus may not act as an N-recognin (Supplementary Fig. 1b).

**Recognition of N-degrons by the ZZ-domain of p62.** In an effort to elucidate the manner by which the ZZ-domain of p62 recognizes N-terminal arginylated BiP/GRP78 (hereafter referred to as R-BiP), we generated the chimeric protein N-terminal R-BiP (R*-E19b-E20b-E21b-D22b; where "*" and subscript "b" represent attachment to the modified N-terminus and BiP residues, respectively) fused to the ZZ-domain using a special expression system (see Methods for details). Using this fusion protein, we determined the complex structure of the ZZ-domain with R-BiP substrate at 1.45 Å resolution (Fig. 1e, Supplementary Fig. 2 and Supplementary Table 2). As expected, the binding site of the ZZ-domain comprises a negatively charged patch for recognition of the positively charged N-terminal $NH_3^+$ group of R-BiP (Fig. 1e). The side chains of Asp129 and Asp149 in the ZZ-domain form hydrogen bonds with the α-amino group of R-BiP (Fig. 1f and Supplementary Fig. 3). Consistent with its important structural role in recognizing the α-amino group of the N-degron, a recent mutagenesis study showed that Asp129 is crucial for functionality of the N-recognin of p62 (see ref.[8]). The carboxylate of Asp176 in the UBR box, which corresponds to Asp129 in p62, was predicted to act as the sole side chain in recognizing the α-amino

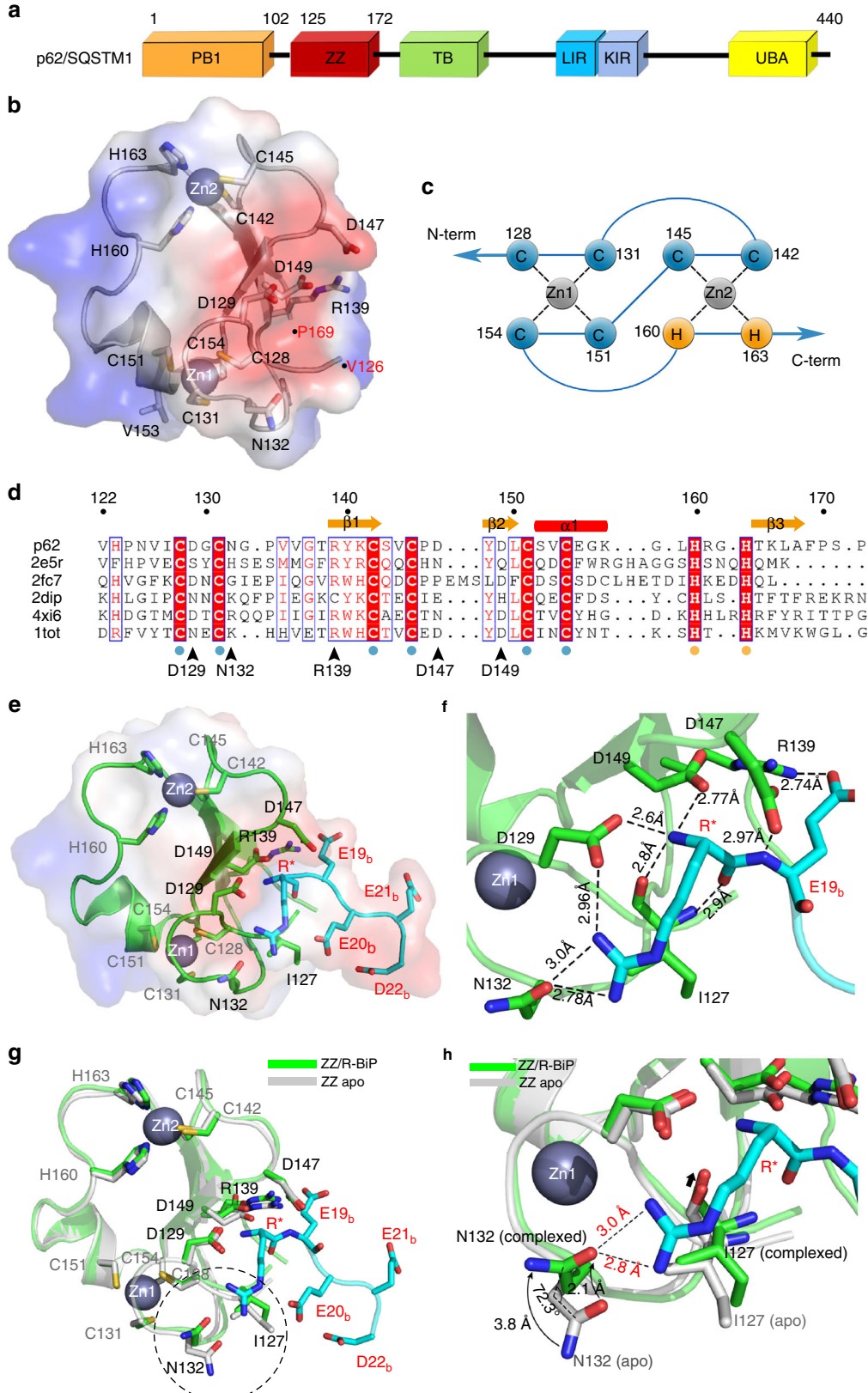

group[31,32,35], however, two side chain carboxylates from Asp129 and Asp149 tightly hold the NH₃⁺ group of the N-degron simultaneously (Fig. 1f). Asp129 is located between two cysteine residues, Cys128 and Cys131 which coordinate the first zinc atom (Zn1), and Asp149 is located between Cys145, which coordinates to the second zinc atom (Zn2), and Cys151, which coordinates to the first zinc (Zn1) (Fig. 1c). Therefore, two zinc atoms are critical for not only stable folding of the ZZ-domain, but also for proper

**Fig. 1** Structure of the ZZ-domain of p62. **a** Domain architecture of p62. The PB1 domain is responsible for oligomerization and localization. The ZZ-domain recognizes both type-1 and type-2 N-degrons. The TB domain, LIR motif and UBA are involved in the interaction with TRAF6, LC3-family proteins and ubiquitin, respectively. **b** Transparent molecular surface showing the electrostatic potential of the ZZ-domain. Negatively and positively charged surfaces are colored red and blue, respectively. Side chains of residues that participate in zinc coordination are shown as stick models and bound zinc ions are shown as slate-colored spheres. The built model of the ZZ-domain comprises residues from Val126 to Pro169 and are marked with dots and labeled. **c** Schematic diagram showing zinc-coordination. The first zinc atom (Zn1) is coordinated by four cysteine residues, and the second zinc (Zn2) by two cysteine and two histidine residues. **d** Sequence alignment of ZZ-domain structures in the Protein Data Bank (2e5r: human α-dystrobrevin; 2fc7: human ZZZ3 protein; 2dip: human SWIM domain containing protein 2; 4xi6: human mind bomb 1; 1tot: mouse CREB-binding protein). Zinc-coordinated residues are strictly conserved among all ZZ-domains, although key residues involved in the recognition of N-degrons (marked with black arrow-heads) are not conserved. **e** Ribbon diagram with transparent electrostatic surface showing the structure of the ZZ-domain in complex with R-BiP substrate (REEED). Residues coordinating zinc atoms and key residues in p62 involved in the recognition of N-degrons are shown as stick models with carbon, nitrogen, and oxygen atoms in green, blue and red, respectively. The bound N-degron is also shown as a stick model with carbon atoms in cyan. Residues of the ZZ-domain are labeled black and those of R-BiP are labeled red with the * and subscript "b" next to the sequence number for clarity. **f** Close-up view of interaction region between the ZZ-domain and R-BiP. Hydrogen bonds are shown as dotted lines and the distance is indicated. **g** Superposition of the structure of apo-ZZ-domain (gray) with that of the R-BiP complex (green). The two structures are almost identical except for Asn132 indicated by a dotted circle. **h** Close-up view of conformational change of Asn132 of the ZZ-domain of p62 upon complex formation

location of the key residues involved in recognizing N-degrons. Moreover, the key carboxylate of Asp129 which is involved in recognizing the α-amino group also forms an ionic interaction with the guanidinium group of the N-terminal arginine residue (Fig. 1f). This completely differs from the UBR box which recognizes positively charged type-1 N-degrons using distantly located negatively charged residues[31]. Another negatively charged residue, Asp147, also participates in the N-degron binding, and in a manner that is not sequence-specific. The side chain carboxylate of Asp147 interacts with the main chain nitrogen atom of the first peptide bond between the first arginine R* and Glu19$_b$ of the N-degron (Fig. 1f). The main chain nitrogen atom of Ile127 also forms a hydrogen bond with the carbonyl oxygen of the first peptide bond of the N-degron (Supplementary Fig. 3).

Since structures of the ZZ-domain have been determined for both the apo and R-BiP complex states, we investigated the possibility of conformational changes in the ZZ-domain of p62 upon complex formation (Fig. 1g). Since the structure is very compact with loops tightly connected by two zinc atoms, no marked conformational changes were identified. However, the side chain of Asn132 is re-oriented to form a specific interaction with the side chain of the N-degron. It is rotated 72.3° and moved by approximately 3.0 Å to facilitate recognition of the guanidinium group of the N-terminal arginine (Fig. 1h). Therefore, the N-terminal arginine residue attached to cleaved BiP by the ATE1 enzyme is recognized by the ZZ-domain of p62 via multiple layers of specificity. As noted in the sequence alignment, this asparagine residue is not conserved in other ZZ-domains (Fig. 1d), and therefore comprises one of the key determinants in addition to the three aspartic acid residues described above.

**Stronger binding of oligomerized p62 to N-degrons**. The dissociation constants ($K_D$) between classic N-recognins and N-degrons (type-1 and type-2) are in the micro-molar range for the recognition of substrates and efficient delivery for degradation by UPS[31,35,36]. As described for the complex structure between the ZZ-domain of p62 and N-degrons, the binding region in the ZZ-domain seems to be very limited (Fig. 1e). The buried surface area upon complex formation is only 546 Å$^2$ and approximately 71% of the surface of primary arginine residues is buried (192 out of 270 Å$^2$), as analyzed by the PISA server[37]. In an effort to determine the binding affinity quantitatively, we measured the $K_D$ value between the ZZ-domain of p62 and R-BiP peptide (REEEDK–FITC) using a fluorescence polarization (FP) method (Fig. 2a). The affinity is extremely weak with a value of over 800

μM (Fig. 2a), as expected from our complex structure, and it is difficult to account for the specific recognition of N-degrons by the ZZ-domain of p62. Intriguingly, the affinity between R-BiP peptide and a GST-fused ZZ-domain is over 5-fold higher (140 μM), which must result from the dimeric effect of GST in solution (Fig. 2a).

p62 is a multi-domain protein (Fig. 1a) and functions in an oligomeric state in the cell. It is reasonable to anticipate that oligomerized p62 should have much higher binding affinity to R-BiP than the monomeric form, as shown in the case of GST-ZZ. The domain responsible for oligomerization is the N-terminal PB1 domain. Therefore, we generated MBP-fused PB1-ZZ (residues 1–181) wild-type (WT) and monomeric mutants K7A and D69A[38]. We generated MBP fusion proteins since it is known that these proteins are highly stable and do not promote protein aggregation in vitro[39]. Following separate purification of WT and mutants, each protein was subjected to size-exclusion chromatography with multi-angle light scattering (SEC-MALS) to ascertain oligomeric states (Fig. 2b). Mutations represented by K7A and D69A in PB1 were sufficient to change the oligomeric state of p62 (Fig. 2b). WT protein formed a large oligomer with molecular mass (MM) of approximately 400 kDa, while the K7A and D69A mutants were mostly observed as monomeric forms with MM of 66 kDa, in addition to a small portion in dimeric form (Fig. 2b). This result indicated that the K7A and D69A mutations in the PB1 domain could disrupt oligomerization[38,40]. Similarly, results of SDS-PAGE and Western blotting with purified WT and D69A mutant were also consistent with the SEC-MALS data (Fig. 2c). Furthermore, in an effort to confirm that oligomerization of p62 is only mediated by the PB1 domain, we performed a small-angle X-ray scattering (SAXS) experiment using dimeric and monomeric species of the D69A mutant (Supplementary Table 3). This result clearly showed that the ZZ-domain is structurally independent from the PB1 oligomerization domain (Supplementary Fig. 4).

To confirm whether the oligomeric state of p62 mediated by PB1 domain affects the recognition of R-BiP, we performed another FP binding assay. The binding affinity of PB1-ZZ WT to R-BiP peptide was over 10-fold higher than that of GST-ZZ (no PB1 domain) as well as monomeric PB1-ZZ mutants (Fig. 2d). The dissociation constant of oligomerized p62 to N-degrons is 10 μM at pH 8.0, which is comparable to that of conventional N-recognins. Indeed, the binding constant itself is not affected upon oligomerization of one component for the 1:1 interaction, although there is enhanced binding affinity as a result of the avidity associated with the multivalent binding sites. Therefore, in

an effort to further confirm the FP results, we performed the $K_D$ measurements using the surface plasmon resonance (SRP) technique. The SPR analysis employing MBP-PB1-ZZ of p62 and R-BiP protein with different combinations showed very interesting results. The $K_D$ values between either p62 WT or D69A mutant and R-BiP were 20.2 and 26.1 µM, respectively, when the p62 protein (WT or mutant) was immobilized onto the sensor chip (Supplementary Fig. 5a, b). However, these values

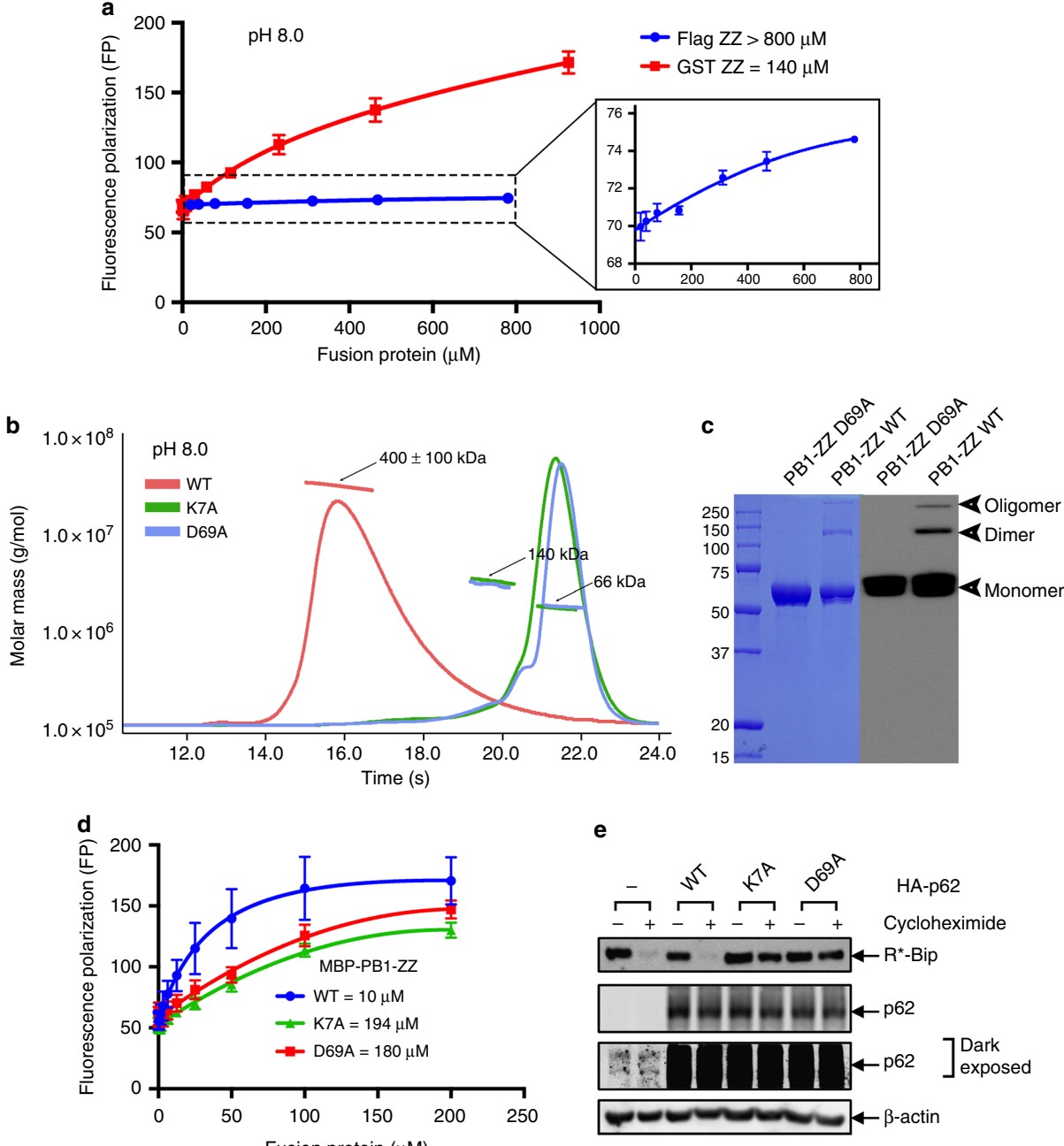

**Fig. 2** Oligomerization of p62 affects the binding affinity and degradation of R-BiP. **a** Binding affinity measurements using FITC-labeled R-BiP peptide against increasing concentrations of the ZZ-domain at pH 8.0. The ZZ-domain fused with dimeric GST (red line) showed higher affinity than that with the flag-tag (blue line), which has extremely weak binding affinity as shown in the inset. The error bars represent standard error of the mean (S.E.M.) of more than three independent experiments. **b** The SEC-MALS results with MBP-PB1-ZZ WT (red line) and mutants K7A (green line) and D69A (sky blue line) at pH 8.0. The horizontal line represents the measured molar mass. Each species is indicated by an arrow with experimental (SEC-MALS) molar mass. WT showed a higher oligomeric state whereas the K7A and D69A mutants mainly adopted a monomeric state with minor dimeric species. **c** The SDS-PAGE results with MBP-PB1-ZZ WT and D69A mutant. The left blue gel is stained with Coomassie Brilliant Blue and the right shows the results of the Western blot. The D69A mutant adopted exclusively a monomeric state whereas WT showed oligomeric forms even under denaturing conditions. **d** Binding affinity measurements using FITC-labeled R-BiP peptide against increasing concentrations of MBP-PB1-ZZ WT (blue line) and mutants K7A (green line) and D69A (red line) at pH 8.0. The error bars represent standard error of the mean (S.E.M.) of more than three independent experiments. **e** Degradation assay of R-BiP generated from Ub–R-BiP using oligomerization defect mutants (K7A and D69A) in HeLa cells in the absence of MG132. Cells were treated with 50 µg/ml cycloheximide, and then subjected to immunoblotting of R-BiP. Oligomerization defect mutants are unable to degrade R-BiP protein in the cell (see also Supplementary Fig. 7 for p62 degradation). Uncropped images of Western blots are shown in Supplementary Figure 11

differ markedly from 42.1 nM and 41.3 µM for p62 WT and D69A mutant, respectively, when the R-BiP protein was immobilized onto the sensor ship (Supplementary Fig. 5c, d). We assumed that the local concentration of immobilized R-BiP is high enough to show extremely tight binding via multivalent interactions with oligomeric p62 WT. To rule out an immobilization effect, we performed isothermal titration calorimetry (ITC) experiments. The $K_D$ values between either p62 WT or D69A mutant and R-BiP peptide were 26.5 and 55.9 µM, respectively, and showed very unusual binding stoichiometry (Supplementary Fig. 6a, b). These binding stoichiometries are difficult to interpret since the exact oligomeric state of p62 WT is unclear and the ITC method might be less useful for interpreting the enhanced binding avidity. Therefore, all subsequent binding affinity measurements were performed with the PB1-ZZ constructs using the FP method. These data can be explained by considering that disruption of oligomerization results in low avidity for the R-BiP substrate and subsequent lack of R-BiP protein degradation in the cell (Fig. 2e and Supplementary Fig. 7).

**Mutational effects of residues for the N-degron recognition.** Since the recognition of substrates by p62 occurs in the cytosol, we decided to compare the binding affinity of mutants using a buffer at pH 8.0. As described in the complex structure, three aspartic acid residues, Asp129, Asp147, and Asp149, and one asparagine residue, Asn132, may play a critical role in substrate recognition. To confirm the role of these residues, we constructed mutants D129N, N132L, D147R, and D149R, and examined the $K_D$ values with R-BiP peptide (Fig. 3a). Clearly, each single point mutation reduced the binding affinity by nearly 30-fold. To further confirm the effect of mutations, the $K_D$ values between either D129N or D147R mutants and R-BiP peptide were measured using the ITC method (Supplementary Fig. 6c, d). The $K_D$ values were 461 and 180 µM for D129N and D147R mutants, respectively, which are quite consistent with the FP results. The basic arginine residue corresponding to Arg139 is also important for the interaction with glutamic acid residue Glu19$_b$ located at the secondary position of N-degrons (Figs. 1f, 3a). It has also been shown that the secondary position of the N-degron partially affects the binding affinity in the UBR box[31,35], and this has been correlated to patients with symptoms of Johanson-Blizzard syndrome[41]. These residues involved in N-degron recognition are strictly conserved in all mammalian p62 proteins, with slight deviation to similar residues in avian, reptile, amphibian and fish proteins (Supplementary Fig. 1a).

To determine whether these residues involved in R-BiP recognition are responsible for the degradation of the R-BiP protein in vivo, we performed cell-based assays using HeLa cells with HA-p62 mutants (D129N, N132L, R139D, D147R and D149R) and Ub-R-BiP[8]. Following DNA transfection, each plate was treated with 50 µg/ml cycloheximide for 12 h (Fig. 3b). Consistent with the in vitro binding assays, all mutants for key determinants showed markedly reduced degradation of R-BiP in vivo (Fig. 3b). The autophagic degradation of R-BiP by these recognition defect mutations resembled that displayed by the oligomerization defect mutations (Fig. 2e).

Since the ZZ-domain of p62 recognizes the R-BiP type-1 N-degron substrate, the key recognition residues of p62 were structurally compared with those of the UBR box (Fig. 3c). Key determinants involved in recognition of the α-amino group are conserved in both N-recognins (Asp129 in p62 and Asp176 in UBR box), but other determinants (Asn132, Arg139, Asp147, and Asp149) completely differ from the UBR box (Fig. 3c). The previously reported mutation D129N of p62 found in patients with neurodegenerative disease[42] can be explained by our data,

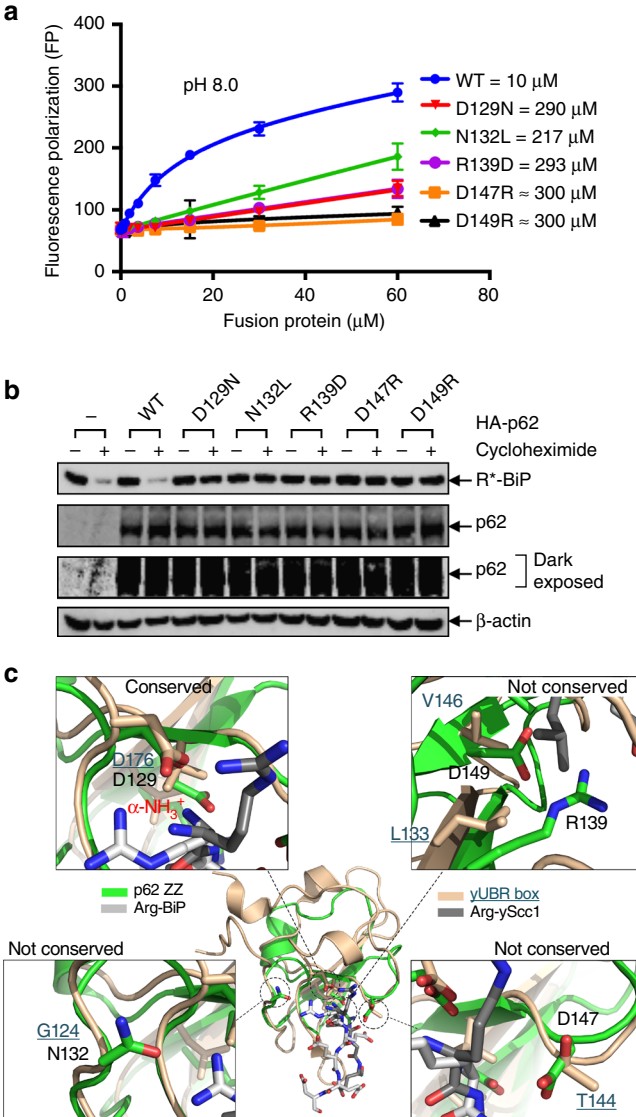

**Fig. 3** Mutational effects of key determinants on the recognition of N-degrons. **a** Binding affinity measurements using FITC-labeled R-BiP peptide against increasing concentrations of p62 mutants (MBP-PB1-ZZ WT—blue line, D129N—red line, N132L—green line, R139D—violet line, D147R—orange line, and D149R—black line) at pH 8.0. The error bars represent standard error of the mean (S.E.M.) of more than three independent experiments. **b** Degradation assay of R-BiP generated from Ub–R-BiP using key determinant mutants (D129N, N132L, R139D, D147R, and D149R) in HeLa cells in the absence of MG132. Cells were treated with 50 µg/ml cycloheximide, and then subjected to immunoblotting of R-BiP. Recognition defect mutants are unable to degrade R-BiP protein in the cell. Uncropped images of Western blots are shown in Supplementary Figure 11. **c** Superposition of structures of R-BiP-bound ZZ-domain (green ribbon) and Scc1-bound UBR box (beige ribbon). Key residues in the ZZ-domain are marked with black dotted circles (center) with a close-up view of each region for details. The labeled residues for the ZZ-domain and UBR box are colored black and dark green (underlined), respectively, for clarity

which might be a consequence of a defect in the recognition of N-degron substrates.

**Both type-1 and type-2 N-degrons recognition.** A classic N-recognin such as Ubr1 possesses two separate domains, a UBR

box and a ClpS-homology domain for the recognition of positively charged type-1 and bulky hydrophobic type-2 N-end rule substrates, respectively[43]. The UBR box utilizes a wider negatively charged pocket than the ZZ-domain of p62, while the ClpS-homology domain utilizes a deeper hydrophobic pocket (Fig. 4a). However, a recent report has shown that the ZZ-domain of p62 also recognizes type-2 N-degrons, although with weaker affinity than with type-1 N-degrons[9]. In an effort to clarify the recognition specificity we measured the $K_D$ values between MBP-PB1-ZZ and various N-degron peptides, including type-1 and type-2 N-degrons (Fig. 4b). As expected, arginine at the primary position showed the strongest binding affinity with, intriguingly, tyrosine and tryptophan residues following in second and third place, respectively. The binding affinity between the ZZ-domain and other type-1 substrates with histidine or lysine residues at the primary position showed ca. a 10-fold reduction, although it was

still significant. Peptides containing proline or glutamic acid residues at the primary position did not interact with the ZZ-domain at all (Fig. 4b). These data clearly explain how the ZZ-domain of p62 recognizes both type-1 and type-2 N-degrons.

However, an understanding of the manner by which the ZZ-domain binds to type-2 substrates is problematic since there is no deep hydrophobic pocket in the ZZ-domain, which is known to be involved in the recognition mode for type-2 N-degrons (Supplementary Fig. 8). Therefore, we determined the structure of the ZZ-domain in complex with a variety of N-degrons comprising three type-1 N-degrons (Fig. 4c and Supplementary Table 2) and five type-2 N-degrons (Fig. 4d and Supplementary Table 4). As described for the R-BiP complex, two aspartic acid residues, Asp129 and Asp149, bind to the α-amino group, and Asp147 forms a hydrogen bond with the first peptide bond, which means that these interactions are conserved in all different N-

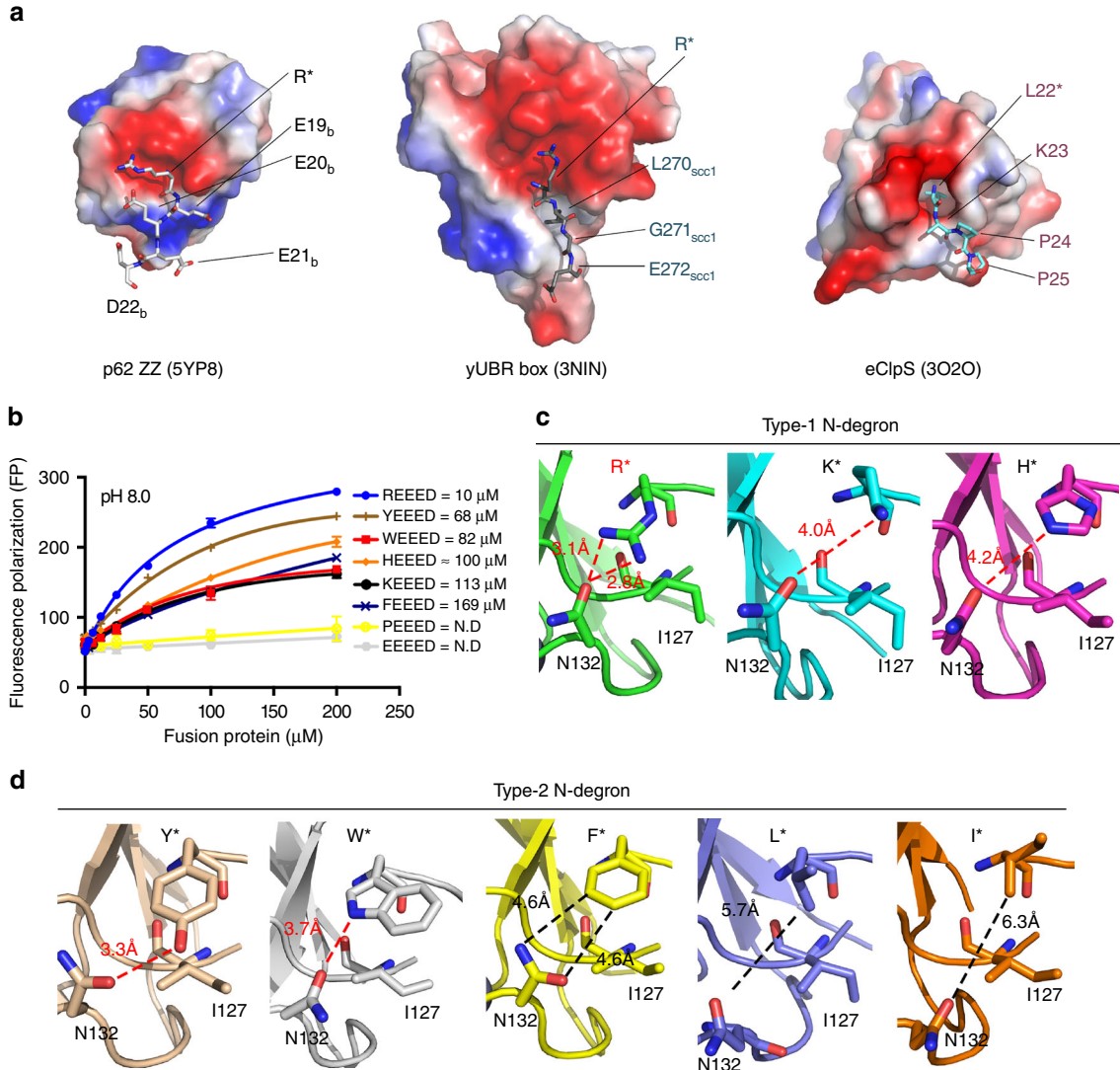

**Fig. 4** Recognition of type-1 and type-2 N-degrons by p62. **a** Molecular surface showing the electrostatic potential of the ZZ-domain (left), yeast UBR box (middle) and *E. coli* ClpS (right). Bound peptides are shown in white, gray, and cyan as stick models for R-BiP, Arg-Scc1 and Leu-peptide substrates, respectively. Red and blue colors represent negatively and positively charged surfaces, respectively (see Supplementary Fig. 8). **b** Binding affinity measurements using various FITC-labeled N-degron peptides against increasing concentrations of MBP-PB1-ZZ at pH 8.0. Different line colors and symbols are used to distinguish each peptide. The error bars represent standard error of the mean (S.E.M.) of more than three independent experiments. **c** Close-up view of the recognition of type-1 N-degrons by the ZZ-domain of p62. **d** Close-up view of the recognition of type-2 N-degrons by the ZZ-domain of p62. The bound peptide and key residues of p62 are shown as stick models. Nitrogen and oxygen atoms are colored blue and red, respectively. Hydrogen bonding and van der Waals contact distances are marked as dashed lines and labeled. Asn132 of p62 is particularly important for the recognition of type-1 as well type-2 N-degron substrates (see Supplementary Fig. 9)

terminal residues. However, the side chain of the N-terminal residue of the N-degron is recognized differently. The side chain of Asn132 which undergoes a conformational change plays a critical role in interacting with the first residue of the N-degron (Supplementary Fig. 9). The bipolar nature of the side chain atoms (O and N) of Asn132 allow for the recognition of positively charged type-1 substrates as well as type-2 substrates, and especially N-terminal tyrosine and tryptophan residues since they possess polar atoms in the side chain (Fig. 4d). The strongest interaction with the arginyl peptide is easily explained by the close bipartite interaction and the hydrogen bonding distance information also provides a rationale for the affinity order (Fig. 4c, d). Furthermore, the hydrophobic side chain of Ile127 guides the orientation of the side chain of the primary residue. Furthermore, the phenyl ring of the phenylalanyl peptide is properly oriented for van der Waals interactions. Therefore, relatively small and branched hydrophobic residues at the primary position might provide very weak (or no) interactions with the binding patch of the ZZ-domain. Our structural information clearly explains the affinity measurement data (Fig. 4b) as well as previous pull-down assay results showing that type-1 N-degrons and only a subset of type-2 N-degron peptides (Phe, Trp, and Tyr) displayed binding affinity with p62 (see ref.[9]).

**pH-dependent oligomerizaion of p62.** A previous UBR box study showed that binding affinity with N-degrons was affected by the protonation state of residues, and that stronger binding was observed at lower pH[31]. Therefore, we examined the dissociation constant of the ZZ-domain of p62 with N-degron at lower pH. The dissociation constant $K_D$ of MBP-PB1-ZZ-domain with R-BiP peptide at pH 6.0 was 338.6 nM, which is an order of magnitude lower than that at pH 8.0 (Fig. 2d and Supplementary Fig. 10). This difference is much more marked than that observed for the UBR box, and most probably results from the protonation states of key side chain residues of the ZZ-domain. To verify this pH effect, we performed the same $K_D$ measurement using a GST-ZZ-only construct, which yielded a $K_D$ value of 11 μM at pH 6.0 (Fig. 5a). Furthermore, the oligomerization defect mutants K7A and D69A showed significantly lower binding affinity with the R-BiP peptide (Fig. 5a), implying that the oligomeric state is affected by pH.

To examine the effect of pH on oligomeric states, MBP-PB1-ZZ WT and mutants K7A and D69A were subjected to SEC-MALS analyses (Fig. 5b). Results showed that the MBP-PB1-ZZ WT polymer is soluble with MW of 1 MDa at pH 6.0 (Fig. 5b), while changes in the size of the mutants at different pH were not significant (Figs. 2b, 5b). Then, we further checked the pH dependency at pH values less than 5.0 (Fig. 5c). The oligomeric states of p62 WT were compared at more physiological (7.4) and acidic (4.5) pH values. The estimated MWs at pH 7.4 and 4.5 are approximately 690 and 180 kDa, respectively. Decameric or higher oligomeric states were observed at pH 7.4, which were larger than those observed at pH 8.0, being hexameric or higher (Fig. 2b). Intriguingly, the oligomeric state of MBP-PB1-ZZ WT at pH 4.5 might be much smaller, such as a trimer. To determine if this small MW is caused by denaturation of p62, a Kratky plot of the SAXS data at pH 4.5 was examined, and clearly showed the pattern of a folded protein (Fig. 5d). To examine the other possibility whether the reducing reagent is important for oligomerization, we performed the same experiments under non-reducing conditions and found that the oligomeric state was not affected by reducing agent. These results clearly showed that p62 protein adopted various sizes (oligomeric states) in a pH-dependent manner.

**pH-dependent regulation of R-BiP aggregates by p62.** As described above, the oligomeric states of p62 mediated by the PB1 domain are affected by the pH conditions, and thus the binding affinity between the ZZ-domain and R-BiP is also markedly influenced. To analyze this phenomenon more systematically, we monitored oligomer (or aggregate) formation of p62 with varying pH (Fig. 6a). The presence of aggregation or high-order oligomer generates an increase in turbidity, which is very similar to the standard chaperone activity assay[44]. As expected, there is no turbidity using p62 WT at neutral pH. However, the turbidity markedly increases from pH 6.0 since p62 forms a polymer with MW of 1 MDa (Fig. 5b). Intriguingly, the turbidity decreased dramatically at more acidic pH, suggesting that the p62 polymer changes to a state comprising smaller oligomers, which is consistent with SEC-MALS results (Fig. 5c). We then employed electron microscopy (EM) to further examine the pH-dependent oligomeric states of p62 (Fig. 6b). It has been shown that the p62 protein forms a filament-like structure using the PB1 domain[30]. Our EM results were extremely intriguing. Most of the proteins were found to adopt huge filamentous forms at pH 6.0 and 5.5, whereas many smaller oligomers with a few filamentous forms were observed at ca. neutral pH (Fig. 6b). At pH 5.0 or below, the oligomeric states of p62 are even lower, and are therefore too small to visualize. Since the oligomerization-defect mutants K7A and D69A are mainly monomers in solution at pH 8.0 and 6.0, they were also too small to visualize using EM.

We further examined the pH-dependent behavior of the R-BiP substrate and clearly this protein aggregated at lower pH values with no recovery whatsoever (Fig. 6c). More interestingly, the mixture between p62 and R-BiP behaved almost identically to that of p62 alone, suggesting that p62 binds to the R-BiP substrate to block aggregation as a chaperone molecule (Fig. 6c). When the pH of the sample mixture decreased to less than 5.0, components in the mixture might be dissociated into smaller sizes of the p62 and R-BiP complex based on the turbidity (Fig. 6c). We also performed EM experiments with a mixture of p62 and R-BiP. In contrast to the EM image of p62 in the presence of ubiquitylated cargos[45], filamentous p62 did not form clusters in the presence of R-BiP. To further examine this phenomenon, the dissociation constant between MBP-PB1-ZZ and R-BiP peptide was measured with varying pH (Fig. 6d). Binding at extreme alkaline (pH 8.5 and 9.0) and acidic (pH 5.0 and 4.5) pH was not detected, and the binding affinity gradually increased from pH 8.0 to 6.0 and then decreased at pH 5.5. The binding affinity near physiological pH is in the micromolar range and increases toward the nanomolar range under slightly acidic conditions near pH 6.0. Ultimately, no binding occurs under lysosomal pH conditions. These are very interesting findings that explain the cellular behavior of p62 in the autophagy pathway from cargo selection to lysosomal degradation.

**Discussion**

The arginyl N-end rule pathway and is mediated by the Ubr1 N-recognin which possesses separate domains involved in the recognition of positively charged type-1 and bulky hydrophobic type-2 N-degrons (Fig. 7a). These domains of classic N-recognins specifically bind target substrates with affinity at the micro-molar level, and the substrates are then ubiquitylated by the C-terminal RING domain[46]. The affinity is optimal for selecting and delivering ubiquitylated substrates into the 26S proteasome. In contrast, the ZZ-domain of p62 can recognize both type-1 and type-2 N-degrons, although its affinity for arginine in the primary residue location is the highest (Figs. 4b, 7a). Our structural and biochemical measurement data showed that the presence of tyrosine or tryptophan at the primary position resulted in relatively

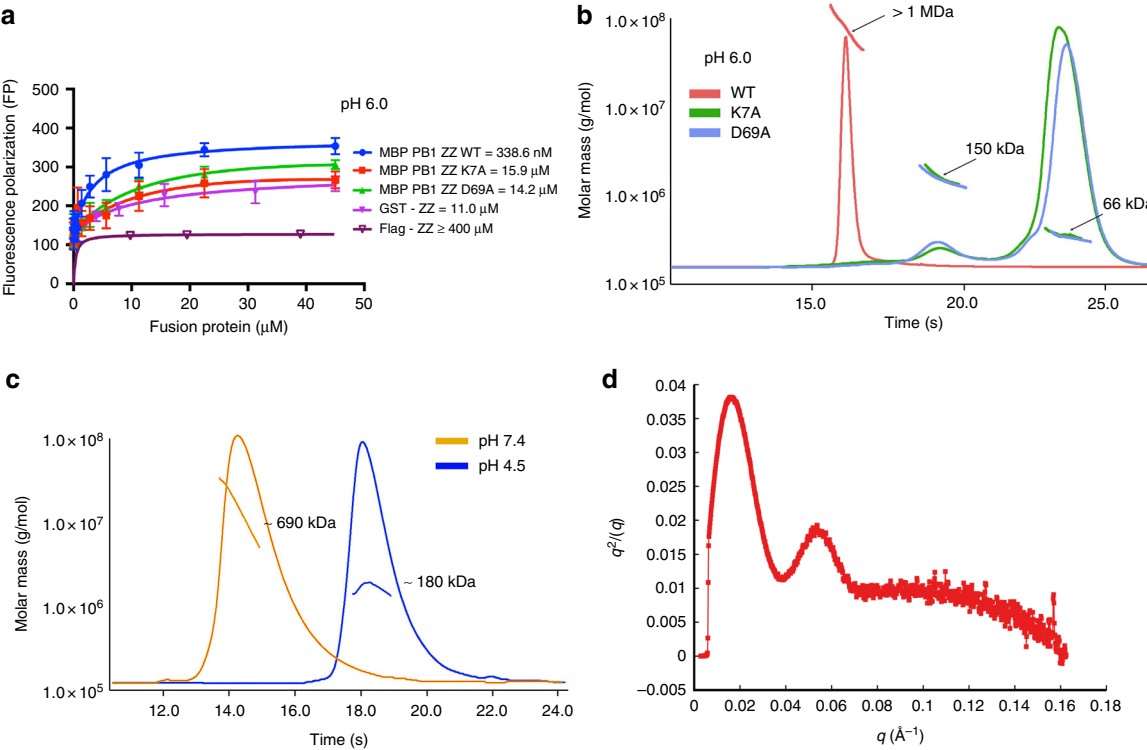

**Fig. 5** Oligomeric states of p62 are controlled by pH. **a** Binding affinity measurements using FITC-labeled R-BiP peptide against increasing concentrations of p62 constructs (MBP-PB1-ZZ WT—blue line, MBP-PB1-ZZ K7A—red line, MBP-PB1-ZZ D69A—green line, GST-ZZ—violet line, and Flag-ZZ—wine line) at pH 6.0. The error bars represent standard error of the mean (S.E.M.) of more than three independent experiments. **b** The SEC-MALS results with MBP-PB1-ZZ WT (red line) and mutants K7A (green line) and D69A (sky blue line) at pH 6.0. The horizontal line represents the measured molar mass. Each species is indicated by an arrow with experimental (SEC-MALS) molar mass. WT protein adopted huge polymeric states whereas the K7A and D69A mutants adopted mainly monomeric states with minor dimeric species as shown in Fig. 2b. **c** The SEC-MALS result with MBP-PB1-ZZ WT at physiological pH 7.4 (orange line) and acidic pH 4.5 (blue line). The horizontal lines represent the measured molar mass, which approximated a decamer at pH 7.4 and trimer at pH 4.5. **d** Kratky plot of SAXS experiment to verify folding of p62 at pH 4.5

high affinity because these side chains contain polar atoms, oxygen in tyrosine and nitrogen in tryptophan that can form hydrogen bonds with the side chain of Asn132 of the ZZ-domain. Except for Asp129 which recognizes the α-amino group of N-degrons (conserved in the UBR box), the other negatively charged residues are not conserved in the UBR box or other ZZ-domains including other autophagy receptors, such as NBR1 (Supplementary Fig. 1b). Intriguingly, the ZZ-domain in plant N-recognin PROTEOLYSIS1 (PRT1) is responsible for recognizing bulky aromatic N-degrons[47,48]. The Asp129 residue is conserved as Asp312 in PRT1, and a few other aspartic acid and metal coordinating cysteine residues are also conserved. Although the Asn132 residue is not conserved, the loops in PRT1 corresponding to those in p62 are slightly longer and hydrophobic residues Val316 and Ile333 are present. Therefore, it is tempting to speculate that the ZZ-domain of PRT1 recognizes bulky aromatic N-degrons in a fashion similar to p62, although specific recognition is derived from the different spatial allocation of recognizing residues.

p62 is not an E3 Ub-ligase and thus there is no step for delivering substrates to the proteasome. Instead, p62 binds to cargo molecules such as protein aggregates and is encapsulated together into the autophagome and is ultimately degraded by lysosome in a suicide manner. Therefore, dissociation of p62 from the cargo molecules is unnecessary for the autophagic pathway. Based on our biochemical data, p62 has extremely low affinity for the R-BiP substrate when present as a monomer, and the functional affinity gradually increases during cellular processes that enhance avidity

via oligomerization. Once it binds, the p62 molecule is degraded together with the protein aggregates in the lysosome (Supplementary Fig. 7). This suicide mechanism is now clearly explained by our biochemical analysis. Furthermore, the monomeric mutants K7A and D69A with altered PB1 domain of p62 are unable to facilitate degradation of R-BiP in cells (Fig. 2e), just as in the case of the binding defect mutants D129N, N132L, R139D, D147R, and D149R with altered ZZ-domain of p62 (Fig. 3b). Although the cellular output of mutants with altered PB1 and ZZ-domain are identical, the underlying mechanisms differ. Oligomerization of PB1 greatly increases its avidity for the R-BiP substrate (Fig. 7b), and a similar situation may exist for the UBA domain, although it is reversed. Although it has been shown that the binding affinity between Ub and UBA is also very weak[49,50], the UBA domain of p62 binds poly- or multi-ubiquitylated substrates very strongly with multiple chances. According to a recent report of the interaction between filamentous p62 and ubiquitylated cargos, they spontaneously coalesce into larger clusters which further interact and crosstalk with autophagy machinery[45].

Our findings on the pH-mediated regulation of p62 oligomerization are intriguing since the pH environment changes as the autophagic pathway progresses[51]. Quantitative analysis by confocal pH-imaging classified the autophagosome (5.8 < pH < 6.2), early autolysosome (5.4 < pH < 5.8), mature autolysosome (5.0 < pH < 5.4) and lysosome (pH < 5.0). Clearly, autophagic flux begins from higher physiological pH of about 7.4 (in mammals) to ultimately a pH below 5.0 within the lysosome. Our in vitro

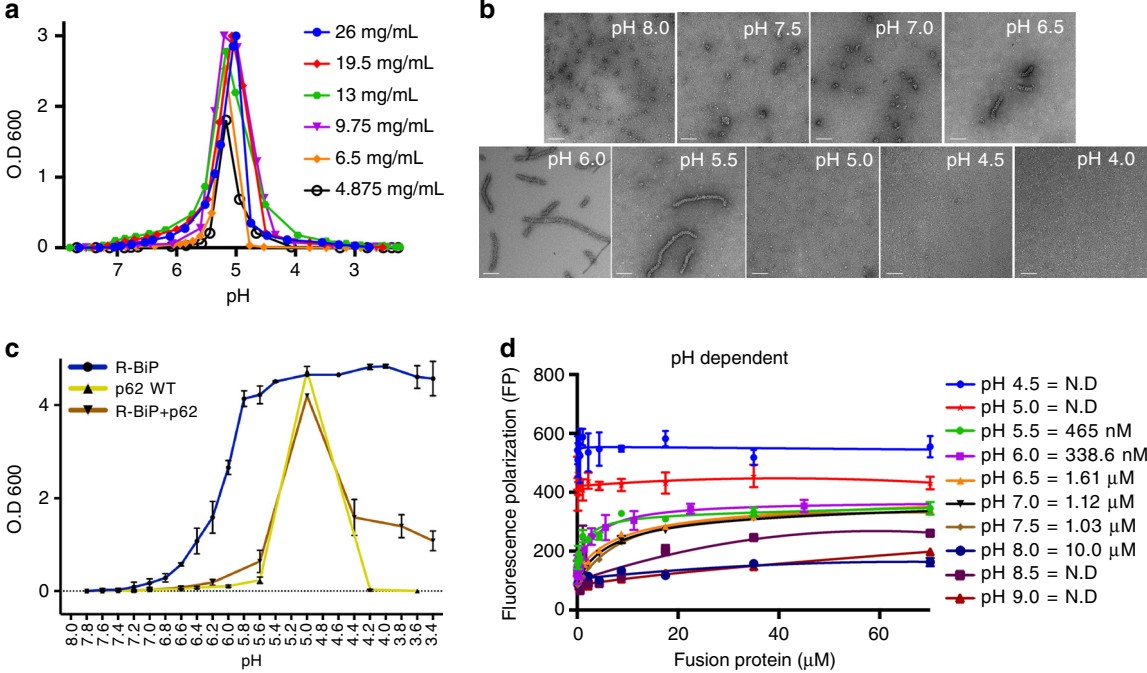

**Fig. 6** pH-dependent assembly and disassembly of p62. **a** Monitoring of high-order oligomerization of p62 by particle turbidity with decreasing pH. Depending on the concentration of MBP-PB1-ZZ, the pH values showing maximal particle size differ slightly. At least three experiments were performed using various protein and HCl concentrations. **b** Representative negative-stain TEM images of MBP-PB1-ZZ at various pH conditions (8.0, 7.5, 7.0, 6.5, 6.0, 5.5, 5.0, 4.5, and 4.0). Filamentous p62 proteins of various lengths are formed at pH 6.5, 6.0 and 5.5. Relatively globular small oligomers are observed at pH 8.0, 7.5 and 7.0. Particle sizes are too small to observe at pH values below 5.0. The indicated scale bar represents 100 nm. **c** Monitoring of aggregation of R-BiP in absence/presence of p62 by particle turbidity with decreasing pH. Although the R-BiP protein is ordinarily denatured at acidic pH, denaturation is limited via protection by the p62 protein. At least three experiments were performed using various protein and HCl concentrations. The error bars represent standard error of the mean (S.E.M.). **d** Binding affinity measurements using FITC-labeled R-BiP peptide against increasing concentrations of p62 at various pH ranging from 4.5 to 9.0. Strong nano-molar scale binding was observed at pH 5.5 and 6.0, while no binding was observed under extremely acidic or basic conditions. The error bars represent standard error of the mean (S.E.M.) of more than three independent experiments

experimental data of p62 can account for the autophagic steps of aggrephagy in the cells. Protein aggregates (with R-BiP) are recognized by small oligomeric p62 with low affinity at pH 7.4. The local concentration of p62 may increase to facilitate further oligomerization since there are more p62 molecules near the aggregates. The interaction between the ubiquitylated cargos and UBA domain of p62 may play a critical role in the formation of a larger and tighter cluster[45]. Furthermore, another autophagy receptor, NBR1, directly cooperates with p62 to form a cluster with greater efficiency. In the meantime, p62 is targeted to the autophagosomal membrane using its LC3-interacting region (LIR) motif. The membrane vicinity might be associated with a relatively low pH due to the negatively charged polar head groups of the lipids. This causes further acceleration of oligomerization, and as a result p62 and cargo aggregates form a very strong complex within the autophagosome, and even stronger complexes within early and mature autolysosomes whose environments are associated with even lower pH. Then, we were interested in examining the fate of the strong complex under acidic pH conditions of the lysosome. It is known that high molecular weight aggregates such as inclusion bodies are very stable within cells since protein aggregates are not easily attacked by proteases. Surprisingly, p62 polymer and aggregates turn into smaller-sized molecules under lysosomal pH conditions, suggesting that the strong complex between p62 and aggregates are now dissociated (Fig. 7b). The smaller proteins, cargo, as well as p62 are now easily degraded by a variety of lysosomal proteases including cathepsins. Although this proposed model needs to be validated

within cells, our findings in the current study provide many insights into the cellular function of the key autophagy receptor p62 with respect to optimal degradation of cargo aggregates, and which broaden our knowledge of N-degron recognition in the N-end rule pathway.

## Methods

**Protein sample preparation.** The PB1-ZZ-domain of p62 (residues 1–181) WT and various mutants were expressed as MBP-fused forms. The mutation was introduced by PCR-mediated site-directed mutagenesis (Supplementary Table 5). The ZZ-domain (residues 122–181) WT and various mutants were expressed as GST-fused forms. Recombinant proteins were overexpressed in *Escherichia coli* BL21(DE3) cells (Novagen, 69450) in LB broth. Cells were grown at 37 °C at 160 rpm until the OD$_{600}$ reached 0.7, and were then immediately induced by addition of isopropyl β-D-1-thiogalactopyranoside (IPTG) to a final concentration of 1 mM. Prior to induction, 200 μM ZnCl$_2$ was added to the culture. Following induction, cells were grown for 16 h at 18 °C. The MBP-fused PB1-ZZ-domains were purified by amylose affinity column chromatography (eluting with 50 mM Tris-HCl pH 8.0, 100 mM NaCl, 1 mM TCEP and 10 mM maltose) and the GST-fused ZZ-domain constructs were purified by GST affinity column chromatography. All constructs were further purified by anion exchange column chromatography using HiTrap Q FastFlow (GE Healthcare, 17-5156-01). Finally, all proteins were passed through a Hi-Load 16/600 Superdex 200 or 16/600 Superdex 75 gel filtration column 75 (GE Healthcare, 28-9893-33) pre-equilibrated with 20 mM Tris-HCl pH 8.0, 150 mM NaCl and 1 mM TCEP.

The domain boundary of ZZ (residue 126–172) was further optimized for better crystallization. For complex structures, various primary residue mutants of N-degron sequence (REEED)-fused ZZ-domains were expressed with an N-terminal His$_6$-LC3B tag. The His$_6$-LC3B-fused ZZ proteins were purified by loading onto a Ni-NTA affinity column and then eluted using a liner gradient of imidazole (0–500 mM). The His$_6$-LC3B tag was removed using human ATG4B protease (Lab made) by overnight incubation at 4 °C, and ZZ constructs with various N-degron sequences were further purified using a cation exchange column. Proteins were

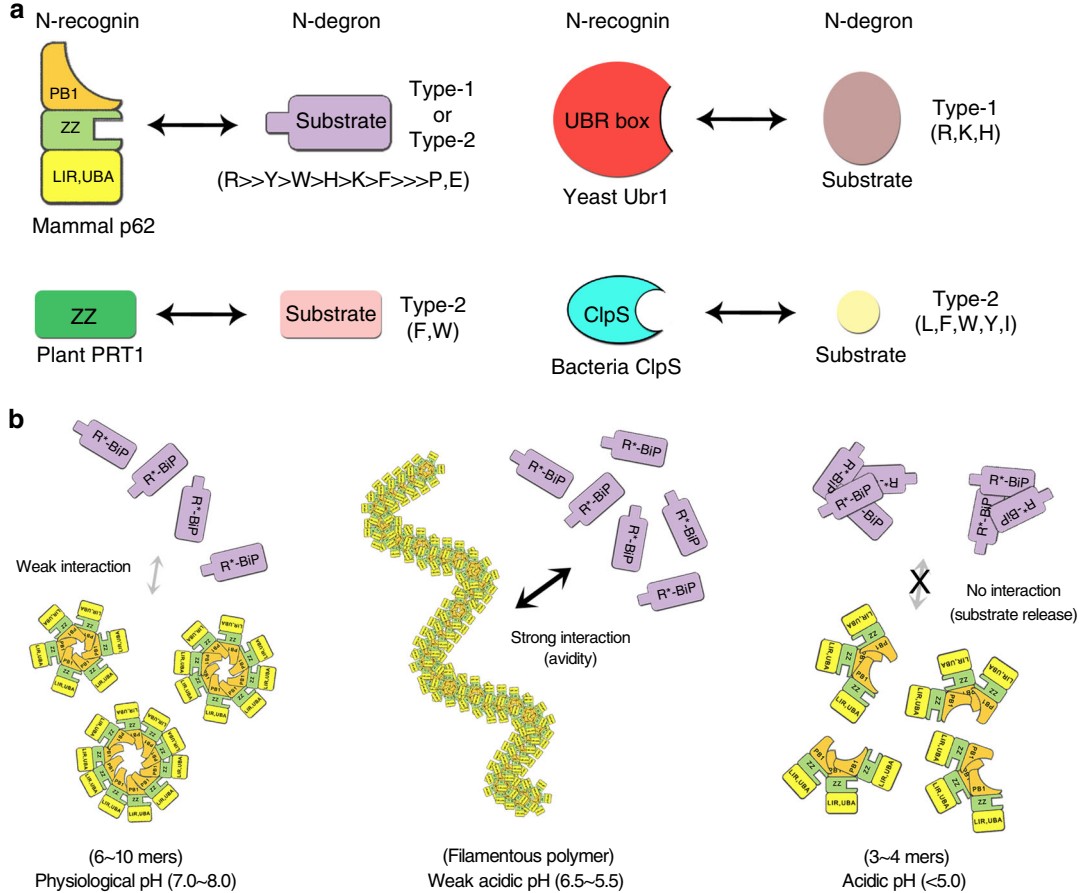

**Fig. 7** Schematic model of N-degron recognition and pH-dependent regulation of p62. **a** The ZZ-domain of mammalian p62 recognizes cargo-bound R-BiP or unknown N-degron (type-1 or type-2) proteins (colored lavender). The PB1, ZZ and remaining (LIR and UBA) domains in p62 are colored orange, light green and yellow, respectively. The ZZ-domain of plant PRT1 E3 Ub ligase recognizes bulky aromatic hydrophobic N-degron. The UBR box from yeast Ubr1 and bacterial ClpS protein recognize basic type-1 and hydrophobic type-2 substrates, respectively. **b** pH-dependent regulation of p62 oligomerization. The R-BiP chaperon (colored lavender) binds to the ubiquitylated aggregate under certain conditions such as stress. The R-BiP containing N-degron is recognized by small-oligomer p62 at physiological pH with very weak affinity (left). p62 forms a long filamentous polymer at lower pH conditions, which might be similar to the environment for forming pre-autophagosomal structures or autophagosomes, and the functional affinity increases markedly via enhanced avidity (middle). The filamentous p62 polymer is converted into smaller-sized oligomers at below pH 5.0, as reflects lysosomal pH conditions, which facilitates release of the substrates from the p62-bound complex (right)

further purified and concentrated to 15~25 mg/ml for crystallization using Hi-Load Superdex 75 pre-equilibrated with 20 mM Tris-HCl pH 8.0, 150 mM NaCl and 1 mM TCEP.

**Crystallization and structure determination**. We crystallized ZZ (residues 122–181 or 126–172) and N-degron-fused ZZ-domains at room temperature in sitting drop plates by the 1:1 mixing of proteins (15–25 mg/ml) and mother liquor (100 mM Bis-Tris pH 6.5 and 20–30% PEG MME 2000—ZZ [122–181]; 100 mM Tris-HCl pH 8.5 and 3.0 M NaCl—ZZ [126–172]; 100 mM MES pH 6.0, 30% PEG 600, 5% PEG 1000 and 10% glycerol—N-degron fused ZZ). Apo and complexed ZZ crystals were flash-frozen in liquid nitrogen with 20–30% glycerol as a cryoprotectant in the original mother liquor. Data were collected at Photon Factory, Spring-8 in Japan and Pohang Accelerator Laboratory (PAL) in South Korea. Initial phases were determined with a 1.77-Å resolution SAD data set using the REEED-fused ZZ crystal collected at the absorption edge of the zinc atom ($\lambda = 1.282282$ Å) at beamline 44XU, Spring-8. Zn-site determination, phasing and automatic model building were performed with the SAD phasing module as implemented in the Phenix software package[52]. The SAD-phased map was of excellent quality, which allowed the AutoBuild utility in Phenix to build a near complete atomic model[53]. Apo ZZ structures were solved by the molecular replacement program Phaser in Phenix[54]. The model solution obtained by Phaser was rebuilt and refined in iterative cycles with Coot[55]. Ramachandran values were calculated with Molprobity[56].

**SEC-MALS**. SEC-MALS experiments were performed using a fast protein liquid chromatography system connected to a Wyatt MiniDAWN TREOS instrument and a Wyatt Optilab rEX differential refractometer. Superdex 200 Increase 10/300 or Superose 6 Increase 10/300 gel filtration columns were pre-equilibrated with three different buffers (50 mM sodium acetate pH 4.5, 50 mM MES pH 6.0, or 50 mM Tris pH 8.0) in the presence of 100 mM NaCl and 1 mM TCEP normalized using ovalbumin and BSA. WT and D69A mutant PB1-ZZ proteins, prepared separately by the methods described earlier, were injected (1–3 mg/ml, 0.5 ml) at a flow rate of 0.5–0.75 ml/min. Data were analyzed using the Zimm model for static light scattering data fitting and represented using an EASI graph with a UV peak in the ASTRA V software (Wyatt).

**Surface plasmon resonance**. All SPR experiments were conducted using a BIA-core 2000 instrument at the Korea Basic Science Institute (KBSI) using a buffer comprising 20 mM HEPES pH 7.5, 100 mM NaCl and 1 mM DTT. Initially, MBP-fused PB1-ZZ WT and D69A mutant were immobilized onto the CM5 chip according to the manufacturer's instructions. Various concentrations of R-BiP N407 (5–100 μM) were then injected at 30 ml/min over the chip. For the converse analysis, the R-BiP protein was immobilized onto the CM5 chip and then various concentrations of either p62 WT or D69A mutant (0.5–50 μM) were used for the experiments. The responses of R-BiP N407 and p62 proteins were calculated by subtracting that of the BSA-immobilized flow cell. All experiments were performed in triplicate. Data were calculated using Scrubber2 software.

**Isothermal titration calorimetry**. For the ITC experiments, ITC buffer (50 mM Tris-HCl pH 8.0, 100 mM NaCl and 1 mM TCEP) was used for the binding experiment. MBP-PB1-ZZ p62 WT, D69A, D129N and D149R mutant proteins

were diluted to a concentration of 20–80 μM in ITC buffer and N-degron peptide (R-E-E-E-D-K) was dissolved in the same buffers at a concentration of 0.5–1.2 mM. The experiment was performed at 25 °C using a Microcal PEAQ-ITC (Malvern). Each peptide was injected 19 times (2 μl each) into 280 μl samples of each protein. The experimental data were calculated using the embedded analyzing software package provided with the instrument. At least three experiments were performed using varied peptide and protein concentrations.

**Small-angle X-ray scattering**. A sample of MBP-PB1-ZZ WT was prepared in gel filtration buffer comprising 25 mM Tris-HCl (pH 8.0), 100 mM NaCl, 1 mM TCEP and 5% (w/v) glycerol. The protein concentration was diluted serially from 20 to 1 mg/ml. Scattering data were collected at beamline 4 °C, PAL, South Korea (Supplementary Tables 3). Briefly, the scattering images from proteins at various concentrations were reduced into 2D data via circular integration. Preliminary analysis of the 2D data with PRIMUS (ATSAS program suite) provided the radius of gyration ($R_g$), Porod volume and experimental molecular weight. Ab initio modeling and averaging of these models were performed using DAMMIF and DAMAVER, respectively[57]. Rigid body modeling of the crystallographic structure on dummy-atom models was computed using the Situs program suite[58].

The initial model employed to perform a molecular simulation against the SAXS envelope was established by combining MBP (PDB ID: 5JST [https://www.rcsb.org/structure/5JST]), p62 PB1 (PDB ID: 4MJS [https://www.rcsb.org/structure/4MJS]), linker (modeled by Chimera) and the ZZ-domain (PDB ID: 5YP7) using the build structure command in Chimera. The SAXS electron envelope map from the ab initio DAMMIN model was generated using the *pdb2vol* command (Situs program suite). The SAXS density map was converted to an MDFF (Molecular Dynamics Flexible Fitting) potential $U_{EM}$ prepared via the MDFF plugin of VMD[59,60]. Rigid body refinement using the *colores* command (Situs) was performed to fit the initial model into the density map. In the first step of MDFF, the g-scale was usually set to 0.3, and in the minimization step, the g-scale was to 10. The MD simulation was typically performed until the system showed no significant change with respect to RMSD (usually over 0.5 ns).

**Cycloheximide-chase protein degradation assay**. HeLa cells were cultured in DMEM (HyClone) containing 10% FBS (HyClone) and 1% penicillin/streptomycin (HyClone). Cells were transiently transfected with plasmids using Lipofectamine 2000 (Invitrogen) for mammalian expression. For protein degradation analysis, HeLa cells at 80% confluence were transiently transfected with plasmids expressing HA-p62 (either WT or mutants) and Ub-R-BiP. For the blocking of protein synthesis, cells were treated with 50 μg/ml cycloheximide (Sigma-Aldrich) for 12 hr prior to cell harvesting. Cultured cells were pelleted by centrifugation and pellets were resuspended in phosphate-buffered saline (PBS). A volume of 150 μl was then mixed with 150 μl of 5X SDS-PAGE loading buffer (125 mM Tris-HCl pH 6.8, 4% SDS, 10% 2-mercaptoethanol and 20% glycerol). Each sample was heated for 5 min and 0.1 mg of total protein was subjected to Western blotting. Following antibodies were used in this study: rabbit monoclonal anti-p62 (Cell Signaling Technology, 8025, 1:1000), rabbit polyclonal anti-R-BiP (Abfrontier, AR02-PA0001, 1:1000), mouse monoclonal anti-β-actin (Sigma-Aldrich, A5441, 1:20,000), rat monoclonal anti-HA (Roche, 1867431, 1:20,000), mouse monoclonal anti-His-HRP (Santa Cruz, sc-8036 HRP, 1:20,000) and mouse monoclonal anti-MBP-HRP (NEB, E8038S, 1:2000).

**Fluorescence polarization assay**. FITC-labeled R-BiP/GRP78 peptide and all mutant peptides were dissolved to 1 mM concentration in buffers (50 mM MES pH 6.0 [or 50 mM Tris pH 8.0], 100 mM NaCl, and 1 mM DTT) and sequentially diluted with binding buffer up to 100 nM in each 40 μL reaction well. Purified GST-ZZ, MBP-PB1-ZZ WT and the respective mutants were also serially diluted in binding buffer and mixed into each reaction well at a concentration ranging from 400 nM to 3 mM. Fluorescent measurements to detect the change in light polarization of the FITC-labeled peptide were performed in a 384-well format on a Corning black plate reader with excitation and emission wavelengths of 485 and 525 nm, respectively. A nonlinear graph of p62 construct concentration-dependent polarization was calculated and drawn using GraphPad Prism 7 software.

**Electron microscopy**. All EM experiments were conducted at KBSI. Purified MBP p62 PB1-ZZ WT protein was diluted to a concentration of 200 nM. Fifty microliters of sample was loaded onto glow-discharged carbon-coated EM grids, and then rinsed and stained with 2% (w/v) uranyl acetate. Images were recorded on a CCD camera (1k/4k, FEI) using a Tecnai G2 field emission gun electron microscope operated at 120 kV with low-dose mode.

**pH-dependent protein aggregation assay**. Purified MBP p62 PB1-ZZ WT and R-BiP N407 proteins were diluted to a concentration of 150 μM and 140 μM, respectively. Two-hundred microliters of each sample was mixed with reaction buffer (50 mM Bis–Tris pH 7.0, 100 mM NaCl and 1 mM TCEP) in a UVette (Eppendorf). After adding 20 μl of 50 mM HCl to the protein sample, the pH and $OD_{600}$ were measured using a semi-micro electrode and UV/VIS spectrometer, respectively.

**Data availability**. Atomic coordinates and structure factor files have been deposited in the Protein Data Bank under following accession codes: 5YP7 (apo), 5YP8 (REEED complex), 5YPA (KEEED complex), 5YPB (HEEED complex), 5YPC (FEEED complex), 5YPE (YEEED complex), 5YPF (WEEED complex), 5YPG (LEEED complex), and 5YPH (IEEED complex). All other data are available from the corresponding author upon reasonable request.

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

## Acknowledgements

We thank the staff at beamline 5C, Pohang Accelerator Laboratory, Korea and beamline BL17A, Photon Factory, Japan for help with the X-ray data collection. This work was in part performed under the International Collaborative Research Program of Institute for Protein Research, Osaka University (ICR-17-05). Diffraction data were collected at the Osaka University beamline BL44XU at SPring-8 (Harima, Japan) (Proposal Nos. 2017A6775 and 2017B6775). We also thank the staff at beamline 4C, Pohang Accelerator Laboratory, Korea, and beamline BL10C, Photon Factory, Japan for help with the SAXS data collection. This work was supported by National Research Foundation grants from the Korean government (NRF-2016R1E1A1A01942623, BRL grant: No. 2015041919, and International Cooperation Program: No. 2015K2A2A6002008).

## Author contributions

D.H.K. made the crystals and solved the structures; D.H.K. and L.K. performed the biochemical experiments; D.H.K., L.K., H.J., and J.H. performed the EM studies; D.H.K. and Y.O.J. performed the SAXS experiments; D.H.K., O.H.P., and Y.P. performed the cell biology experiments; D.H.K., J.H., Y.K.K., and H.K.S. analyzed the data; D.H.K. and H.K. S. designed the experiments and wrote the manuscript.

## Additional information

**Competing interests:** The authors declare no competing interests.

