## [Peer Review File · Nature Communications]

Reviewers' comments:

Reviewer #1 (Remarks to the Author):

The manuscript by Kwon et al. characterized the physical interactions of p62 ZZ domain with various type-1 and -2 N-degrons using X-ray crystallography together with some biophysical methods and revealed the structure of p62 ZZ and its recognition mode of N-degrons at atomic resolution. Moreover, the authors found that the oligomerization state of p62 was markedly affected by pH in vitro, which regulated the binding affinity of p62 ZZ to N-degrons. Based on the results, the authors proposed that p62 mediates autophagic degradation of N-degrons by changing both self-oligomerization and binding affinity to N-degrons along with the steps of autophagy. Throughout the manuscript, structural studies are fine and clearly explain how p62 ZZ recognizes different types of N-degrons. Characterization of the oligomerization state of p62 by SEC-MALS is also effectively performed. These new knowledges will be beneficial for the field. However, there are some critical concerns in the proposed model and in some experiments described below, which must be resolved prior to be published in Nature Communications.

Major points

1) The authors attributed the effect of pH on the affinity solely to the oligomerization state of p62. However, there are some contradictory results. For example, the authors showed that K7A or D69A mutation made p62 into mainly monomers, which was not largely affected by pH difference (pH 6 vs 8 in Fig. 5b and 2b). However, the K_d value was markedly affected by pH difference ($K_d = 3.3$ (pH6) and $194\mu\text{M}$ (pH8) for K7A and 3.2 (pH6) and 180 uM (pH8) for D69A). The K_d values ($3.3\mu\text{M}$ and $3.2\mu\text{M}$) of K7A and D69A at pH6 are much smaller (stronger) than GST-ZZ ($K_d = 140\mu\text{M}$ at pH8) although the mutants are mainly monomers whereas GST-fusion is mainly dimers. These data suggest that pH directly affects the affinity between p62 and N-degrons other than the oligomerization state of p62.

Moreover, the proposed model that oligomerization of p62 enhances its affinity to N-degrons cannot be understood logically. In the case of recognizing poly-Ub chains by UBA, oligomerization of p62 would enhance the affinity with poly-Ub chains through multivalent polyUBA-polyUb interactions. However, in the case of recognizing N-degrons by ZZ, oligomerization of p62 would not enhance the affinity because N-degron is not multivalent. Each ZZ in the oligomerized p62 will interact with each N-degron in a 1:1 manner, so oligomerization should not affect the K_d value unless N-degrons are also oligomerized. Reconsider the logic throughout the manuscript and perform additional experiments that support the new logic.

2) Throughout the manuscript, the affinity between p62 and N-degrons was studied by FP method except for Supplementary Fig. 4, where SPR was used. The FP method is convenient for checking rough affinity, but not so accurate. The sole SPR experiment was performed using R*-19~407, which is totally different from the peptides used for the FP method. Perform SPR or ITC using the same constructs used for the FP method and re-check the affinity at least for some critical data.

3) The authors consider that the interaction of p62 with N-degrons is solely mediated by the ZZ domain in this manuscript. However, Cha-Molstad et al (Nat Commun 2017) proposed that not only ZZ, but also PB1 of p62 contribute to the interaction with N-degrons. In this manuscript, the authors did not show any data supporting that N-degron recognition is solely mediated by ZZ, and considering the observation described in 1), it is reasonable to think that PB1 domain (especially oligomerized one) also contributes to the high affinity with N-degrons other than causing oligomerization. Study the direct contribution of the PB1 domain to the affinity with N-degrons.

4) Based on the in vitro oligomerization state of p62, the authors proposed a model of p62 function in cells. However, p62 interacts with various factors in cells and the oligomerized state of p62 seems to be affected by such interactions. For example, a recent manuscript (Zaffagnini et al. EMBO J 2018) proposed that ubiquitinated proteins form large clusters together with p62. Therefore, the model solely based on the purified p62 experiments in vitro is too biased to propose the functions in cells. Remove Fig. 7b and rewrite discussion in a non-biased way.

5) In vivo data in Fig. 2e and 3b are confusing. In cells without HA-p62, degradation of R*-BiP was observed. Is this degradation mediated by endogenous p62? If so, why expression of HA-p62

mutants impaired the degradation of R*-BiP? Did they function as dominant-negative? Moreover, why didn't the level of p62 change at all? If the degradation of R*-BiP is dependent on p62-mediated autophagy, then the level of p62 would also be reduced together with R*-BiP. From this experiment, we cannot know whether the degradation of R*-BiP is dependent on autophagy. Perform control experiment under autophagy-inhibited conditions.

Minor points

- 1) The authors used fusion protein between p62 ZZ and N-degrons for crystallography, but there is no description on the linker region in the results section. Describe whether the electron density of the linker was visible or not, and if not, indicate the linker region using a broken line in Fig. 1e. Provide reasonable discussion on the little effect of linkers on the complex structure (if there is a possibility of the effect of linker on the structure, describe it).
- 2) In the discussion using Fig 4c and 4d, the location of Asn132 and N-degron is very important. Provide electron density map for each N-degron and Asn132 as supplementary data and reveal that their structures have been unambiguously determined.
- 3) In page 6, line 115, "Fib. 1b" should be "Supplementary Fig. 1b".
- 4) In page 7, line 130, "Asp129 was predicted to ... in the case of a UBR box". Asp129 is a number for p62 and so the sentence is confusing. The authors should write as "AspXXX in a UBR box, which corresponds to Asp129 in p62, was predicted to...".
- 5) In page 8, lines 168-169, "Although the affinity is ..., it is difficult to account for ...". This sentence is difficult to understand. Why "Although"?

Reviewer #2 (Remarks to the Author):

From the work of Cha-Molstad et al. published in two recent papers in Nat. Cell Biol. 2015 (PMID: 26075355) and in Nat. Commun. 2017 (PMID: 28740232) it was shown that N-terminal arginylated BiP (GRP78) is recognized by p62/SQSTM1, assembled into p62 bodies by polymerized p62, and degraded with p62 by autophagy, and that this is mediated by binding of the arginylated BiP to the ZZ domain of p62. So, p62 acts as an N1-recognin in the N-end rule pathway to direct both type 1 and type 2 N-degrons to autophagic degradation.

In the present manuscript with the title "Insights into degradation mechanism of N-end rule substrates by p62/SQSTM1 autophagy adaptor» the authors have solved the structure of the ZZ zinc-finger domain of p62/SQSTM1 alone and in complex with several short peptides representing type 1 and type 2 N-degrons and elucidate the binding mode and affinities involved. The positive and important contribution of p62's ability to polymerize via the PB1 domain to increase the binding affinity is clearly demonstrated. A very interesting and surprising discovery reported here is that polymerization and disassembly of p62 polymers are pH-dependent. Hence, pH may be a very important local determinant of the behavior of p62 as an autophagy receptor along the autophagic pathway from cargo recognition and autophagosome formation to autolysosome formation and degradation of the cargo, and p62 itself.

The novelty of this work lies first and foremost in the finding of a pH dependent polymerization of p62. This is an intriguing finding. The elucidation of the binding mode of N-degron peptides to the ZZ domain is also an important step forward in our understanding of cargo recognition of p62 that does not involve the UBA domain and ubiquitinated cargo. However, as mentioned above, it was already known that the N-degrons bound to the ZZ domain compromising somewhat the novelty of this main part of the work.

Major points

1. The authors state in line 205-206 that: "Since the recognition of substrates by p62 occurs in the cytosol, we decided to compare the binding affinity of mutants using a buffer at pH 8.0. ". I find it strange that the authors did not use a buffer of pH 7.4 instead. This would have been the physiologically relevant pH to choose to study the interactions occurring in the cytosol. I am

tempted to go as far as to insist that the experiments done at pH 8.0 should be done at pH 7.4 in a revised version of this paper.

2. The turbidity assay used needs more validation by testing the ability to form filaments by negative stain TEM at some of the most important pH steps used like 7.5, 7.0, 6.5, 6.0, 5.0 and 4.0. Although TEM at 6.0 is shown in Fig. 5 it should be included as part of a series verifying the turbidity assay.

Minor points

3. It would be interesting to see negative stain TEM of the R-BiP:p62 complex at some selected pH values.

4. Are the authors able to rationalize the pH dependence of p62 polymerization by an analysis of the protonation states of critical residues involved in the electrostatic interactions crucial for PB1-PB1 interactions? Such residues are K7, R21, D69 and E82.

5. Comparing Fig 2a and Fig 5a it is clear that in Fig 5a the authors should include FP binding studies with Flag-ZZ at pH 6.0 to control that the pH effect on binding to GST-ZZ is mediated by the actual binding between the degron peptide and the ZZ domain and not an indirect effect mediated by pH effects on GST.

6. The term oligomer is used throughout for polymers of p62. When the MW is around 1 MDa (lines 314-315) it is more relevant to use the term polymer rather than oligomer as there must be more than ten p62 molecules forming a 1 MDa polymer.

7. Ref 8 should be added in line 78 following "...is recognized by the ZZ domain of p62;"

8. Line 161: delete "approximately"

9. Line 186: Ref 38 should be added along with ref 40

10. Line 283: "higher" should be corrected to "lower"

Revisions to the manuscript and point-by-point responses to the comments of the reviewers

Overview

- We have now performed several more experiments including ITC and SPR for additional binding constant measurements, negative-stained EM, and the monitoring of p62 (and R-BiP) degradation in cells. The rationale for the high avidity has now been clearly described. All additional experiments suggested by the reviewers have proven to be very constructive and we have re-written the text of the manuscript according to the reviewers' suggestions. Detailed responses to the reviewer comments follow.

Responses to Referee #1

<< *The manuscript by Kwon et al. characterized the physical interactions of p62 ZZ domain with various type-1 and -2 N-degrons using X-ray crystallography together with some biophysical methods and revealed the structure of p62 ZZ and its recognition mode of N-degrons at atomic resolution. Moreover, the authors found that the oligomerization state of p62 was markedly affected by pH in vitro, which regulated the binding affinity of p62 ZZ to N-degrons. Based on the results, the authors proposed that p62 mediates autophagic degradation of N-degrons by changing both self-oligomerization and binding affinity to N-degrons along with the steps of autophagy. Throughout the manuscript, structural studies are fine and clearly explain how p62 ZZ recognizes different types of N-degrons. Characterization of the oligomerization state of p62 by SEC-MALS is also effectively performed. These new knowledges will be beneficial for the field. However, there are some critical concerns in the proposed model and in some experiments described below, which must be resolved prior to be published in Nature Communications. >>*

<< *Major points*

1) *The authors attributed the effect of pH on the affinity solely to the oligomerization state of p62. However, there are some contradictory results. For example, the authors showed that K7A or D69A mutation made p62 into mainly monomers, which was not largely affected by pH difference (pH 6 vs 8 in Fig. 5b and 2b). However, the K_d value was markedly affected by pH difference (K_d = 3.3 (pH6) and 194 μM (pH8) for K7A and 3.2 (pH6) and 180 μM (pH8) for D69A). The K_d values (3.3 μM and 3.2 μM) of K7A and D69A at pH6 are much smaller*

(stronger) than GST-ZZ ($K_d=140\mu\text{M}$ at pH8) although the mutants are mainly monomers whereas GST-fusion is mainly dimers. These data suggest that pH directly affects the affinity between p62 and N-degrons other than the oligomerization state of p62.

- Yes, we previously found that the pH directly affects the affinity between the UBR box and N-degrons (Choi *et al.*, 2010), which must be mediated by the protonation states of key interacting residues including the α -amino group. As described in the manuscript, the pH effect on binding affinity between the ZZ-domain of p62 and N-degron is more significant than that between the UBR box and N-degron (see pp. 13~14, “pH-dependent oligomerization of p62” section in the revised manuscript).

In an effort to address the comments regarding the binding affinity values for monomeric mutants K7A and D69A, we performed the fluorescence polarization (FP) assay again (this time filtering the newly prepared sample before the fluorescence measurements) and revised the K_D values, which are now more reasonable (please see Fig. 5a in the revised manuscript). The K_D values of K7A and D69A (15.9 and 14.2 μM , respectively) at pH 6.0 are weaker than that of GST-ZZ (11.0 μM). One thing we have to note is that monomeric mutants of p62 possess a minor portion of dimeric species that affect the binding affinity measurements.

<< Moreover, the proposed model that oligomerization of p62 enhances its affinity to N-degrons cannot be understood logically. In the case of recognizing poly-Ub chains by UBA, oligomerization of p62 would enhance the affinity with poly-Ub chains through multivalent polyUBA-polyUb interactions. However, in the case of recognizing N-degrons by ZZ, oligomerization of p62 would not enhance the affinity because N-degron is not multivalent. Each ZZ in the oligomerized p62 will interact with each N-degron in a 1:1 manner, so oligomerization should not affect the K_d value unless N-degrons are also oligomerized. Reconsider the logic throughout the manuscript and perform additional experiments that support the new logic. >>

- We greatly appreciate the comments on the issue regarding affinity between individual monomers and avidity (functional affinity) between oligomers with multiple binding sites and monomer. Yes, there is no affinity change if the interaction itself is comprised of a 1:1 manner, however, the functional affinity varies. The oligomer possesses multiple

binding sites and the monomeric N-degron binds to oligomer with strong avidity. In general, the apparent dissociation constants reflect tight binding. Therefore, it is now clear that the binding affinity itself is directly affected by pH and that the pH markedly changes the oligomeric states of p62, which in turn affects functional affinity. With this in mind, the text has been revised in many parts. We hope the issue on the enhanced binding affinity of 1:1 interaction is now clearly explained by avidity (or functional affinity) term, which was not introduced at all in the previous manuscript.

<< 2) Throughout the manuscript, the affinity between p62 and N-degrons was studied by FP method except for Supplementary Fig. 4, where SPR was used. The FP method is convenient for checking rough affinity, but not so accurate. The sole SPR experiment was performed using R-19~407, which is totally different from the peptides used for the FP method. Perform SPR or ITC using the same constructs used for the FP method and re-check the affinity at least for some critical data. >>*

- As reviewer #1 suggested, we performed the SPR experiments using immobilized p62 WT (or D69A mutant) with R-BiP protein as an analyte, in addition to performing the same experiments using immobilized R-BiP with WT or D69A mutant as analytes. The experimental data are now included as Supplementary Fig. 5 in the revised manuscript. The K_D values are quite comparable with the FP results if the p62 proteins are immobilized on the sensor chip (Supplementary Fig. 5a, b). However, extremely tight binding was observed if R-BiP is immobilized on the sensor ship (Supplementary Fig. 5c). This may result from the immobilized R-BiP acting as a multimeric substrate with oligomeric p62 WT binding extremely tightly. Consistent with this, it was found that monomeric D69A binds to immobilized R-BiP with much weaker affinity (Supplementary Fig. 5d).

Furthermore, we also performed ITC experiments using WT, D69A, D129N and D147R mutants with R-BiP peptide as a titrant. These data are included as Supplementary Fig. 6 in the revised manuscript. The ITC technique turned out to be a very good technique for dissecting the differences between the oligomerization and binding defect mutants. The K_D values for WT and D69A mutant are quite similar to the SPR results, however, the binding stoichiometries are very unusual (Supplementary Fig. 6a, b), which are 32.2:1 and 19.2:1 for WT and D69A mutant, respectively. If the

stoichiometry was fixed to several lower values as a test, the K_D values were changed or obtained with unreasonable fitting of the data by the software. Interestingly, the binding constant measurements with mutants of N-degron recognition showed clear results and are consistent with the FP data (Supplementary Fig. 6c, d). We have described these data in our revised manuscript (p. 9~10, latter part of the section “Oligomerization of p62 is required for stronger binding to N-degrons”).

<< 3) The authors consider that the interaction of p62 with N-degrons is solely mediated by the ZZ domain in this manuscript. However, Cha-Molstad et al (Nat Commun 2017) proposed that not only ZZ, but also PB1 of p62 contribute to the interaction with N-degrons. In this manuscript, the authors did not show any data supporting that N-degron recognition is solely mediated by ZZ, and considering the observation described in 1), it is reasonable to think that PB1 domain (especially oligomerized one) also contributes to the high affinity with N-degrons other than causing oligomerization. Study the direct contribution of the PB1 domain to the affinity with N-degrons. >>

- To examine whether the PB1 domain or the linker between PB1 and the ZZ domain participate in N-degron binding, we performed FP experiments using MBP-PB1 and MBP-PB1-linker constructs. FITC-labelled R-BiP peptide and different length constructs of proteins (MBP-PB1 [1-102], MBP-PB1-Linker [1-121], and MBP-PB1-Linker-ZZ [1-181]) in 50 mM Tris pH 8.0 buffer were used for K_D measurement (The details are the same as the other FP assays, which are described in “Fluorescence polarization assay” in Methods section).

As shown above, there is no contribution by either of the two regions to the affinity with N-degron (N.D: not determined). Therefore, only ZZ-domain of p62 is responsible for the recognition of N-degron.

<< 4) *Based on the in vitro oligomerization state of p62, the authors proposed a model of p62 function in cells. However, p62 interacts with various factors in cells and the oligomerized state of p62 seems to be affected by such interactions. For example, a recent manuscript (Zaffagnini et al. EMBO J 2018) proposed that ubiquitinated proteins form large clusters together with p62. Therefore, the model solely based on the purified p62 experiments in vitro is too biased to propose the functions in cells. Remove Fig. 7b and rewrite discussion in a non-biased way. >>*

- We initially thought that original Fig. 7b might provide some insight into the pH-dependent regulation of p62 for autophagy researchers in a graphical manner, however, as pointed out by the reviewer, we realized that given the complexity of the autophagic pathway, it could not be explained only by considering the pH changes shown in previously presented Fig. 7b. Therefore, according to the reviewer suggestion, original Fig. 7b has almost been completely removed and revised as our new Fig. 7b in the revised manuscript. This figure shows the assembly and disassembly of p62 depending on pH conditions that we found in our *in vitro* experiments. Revision of Fig. 7b was decided upon *in lieu* of its complete removal to facilitate better discussion. A very recent paper (Zaffagnini *et al.*) introduced by the reviewer contains a plethora of valuable data for discussion, and in particular the finding that ubiquitylated cargos mediated p62 clustering has been referred to in several parts of the Discussion section.

<< 5) *In vivo data in Fig. 2e and 3b are confusing. In cells without HA-p62, degradation of R*-BiP was observed. Is this degradation mediated by endogenous p62? If so, why expression of HA-p62 mutants impaired the degradation of R*-BiP? Did they function as dominant-negative? Moreover, why didn't the level of p62 change at all? If the degradation of R*-BiP is dependent on p62-mediated autophagy, then the level of p62 would also be reduced together with R*-BiP. From this experiment, we cannot know whether the degradation of R*-BiP is dependent on autophagy. Perform control experiment under autophagy-inhibited conditions.>>*

- The previous data have been generated using an excess amount of HA-p62 DNA (WT or mutants) to highlight the degradation of R-BiP proteins in cells. Yes, the R-BiP protein is co-degraded with p62 *via* the autophagy pathway. To clarify this, we have performed the same experiments using reduced amounts of HA-p62 DNA (approximately 1/6) and longer cycloheximide treatments. As shown in Supplementary Fig. 7 in the revised manuscript, HA-p62 WT degradation with R-BiP substrate can be clearly observed. Although HA-p62 mutants, comprising oligomerization defect D69A and N-degron recognition defect D129N, are also degraded with R-BiP slightly, the R-BiP substrate is not degraded, in contrast to the case of WT.

<< *Minor points*

1) *The authors used fusion protein between p62 ZZ and N-degrons for crystallography, but there is no description on the linker region in the results section. Describe whether the electron density of the linker was visible or not, and if not, indicate the linker region using a broken line in Fig. 1e. Provide reasonable discussion on the little effect of linkers on the complex structure (if there is a possibility of the effect of linker on the structure, describe it).>>*

- For clarity, we describe the exact sequence number of the construct in the Methods section, and have also included Supplementary Fig. 2 in the revised manuscript. This figure shows the N-terminal amino acid sequence of the construct, crystal packing, and electron density map of the linker region. The N-degron residues protrude from the ZZ-domain core and bind to the neighboring symmetry-equivalent molecule. The electron density map for this region is sufficiently clear for model building and there is no invisible region. We included the R-EEED complex structure only in the revised manuscript (Supplementary Fig. 2) and all the other complex structures are similar as shown below (next page). To avoid redundancy, they have not been included in the revised manuscript.

<< 2) In the discussion using Fig 4c and 4d, the location of Asn132 and N-degron is very important. Provide electron density map for each N-degron and Asn132 as supplementary data and reveal that their structures have been unambiguously determined. >>

- **The electron density maps for the side chain of Asn132 in all different complex structure are clear, although the resolution limits of each data set differ. Supplementary Fig. 9 is now included in the revised manuscript which shows the electron density map of the ZZ-domain recognizing eight different N-degrons.**

<< 3) In page 6, line 115, “Fib. 1b” should be “Supplementary Fig. 1b”.>>

- **This has been modified to read “Supplementary Fig. 1b”. Thank you.**

<< 4) In page 7, line 130, “Asp129 was predicted to ... in the case of a UBR box”. Asp129 is a number for p62 and so the sentence is confusing. The authors should write as “AspXXX in a UBR box, which corresponds to Asp129 in p62, was predicted to...”.>>>

- **We appreciate the detailed comment and have corrected the sentence as suggested in the revised manuscript.**

<< 5) In page 8, lines 168-169, “Although the affinity is ..., it is difficult to account for ...”. This sentence is difficult to understand. Why “Although”? >>

- **Thank you. “Although” has been removed and the sentence has been modified to read “The affinity is extremely weak with a value of over 800 μ M (Fig. 2a), as expected from our complex structure, and it is difficult to...” in the revised manuscript.**

Responses to Referee #2

<< From the work of Cha-Molstad et al. published in two recent papers in Nat. Cell Biol. 2015 (PMID: 26075355) and in Nat. Commun. 2017 (PMID: 28740232) it was shown that N-terminal arginylated BiP (GRP78) is recognized by p62/SQSTM1, assembled into p62 bodies by polymerized p62, and degraded with p62 by autophagy, and that this is mediated by

binding of the arginylated BiP to the ZZ domain of p62. So, p62 acts as an N1-recognin in the N-end rule pathway to direct both type 1 and type 2 N-degrons to autophagic degradation. In the present manuscript with the title “Insights into degradation mechanism of N-end rule substrates by p62/SQSTM1 autophagy adaptor» the authors have solved the structure of the ZZ zinc-finger domain of p62/SQSTM1 alone and in complex with several short peptides representing type 1 and type 2 N-degrons and elucidate the binding mode and affinities involved. The positive and important contribution of p62’s ability to polymerize via the PB1 domain to increase the binding affinity is clearly demonstrated. A very interesting and surprising discovery reported here is that polymerization and disassembly of p62 polymers are pH-dependent. Hence, pH may be a very important local determinant of the behavior of p62 as an autophagy receptor along the autophagic pathway from cargo recognition and autophagosome formation to autolysosome formation and degradation of the cargo, and p62 itself.

The novelty of this work lies first and foremost in the finding of a pH dependent polymerization of p62. This is an intriguing finding. The elucidation of the binding mode of N-degron peptides to the ZZ domain is also an important step forward in our understanding of cargo recognition of p62 that does not involve the UBA domain and ubiquitinated cargo. However, as mentioned above, it was already known that the N-degrons bound to the ZZ domain compromising somewhat the novelty of this main part of the work. >>

<< Major points

1. The authors state in line 205-206 that: “Since the recognition of substrates by p62 occurs in the cytosol, we decided to compare the binding affinity of mutants using a buffer at pH 8.0. “. I find it strange that the authors did not use a buffer of pH 7.4 instead. This would have been the physiologically relevant pH to choose to study the interactions occurring in the cytosol. I am tempted to go as far as to insist that the experiments done at pH 8.0 should be done at pH 7.4 in a revised version of this paper. >>

- We performed several experiments using buffer at more physiological pH (7.4). Please refer to the SEC-MALS experiments for examination of the oligomeric state (Fig. 5c in the revised manuscript), and the EM image for visualization of the oligomer at pH 7.5. Binding affinity measurements were also performed at pH 7.4 (data not included) and

the K_D value was nearly the same as at pH 7.5 (Fig. 6d in the revised manuscript).

<< 2. The turbidity assay used needs more validation by testing the ability to form filaments by negative stain TEM at some of the most important pH steps used like 7.5, 7.0, 6.5, 6.0, 5.0 and 4.0. Although TEM at 6.0 is shown in Fig. 5 it should be included as part of a series verifying the turbidity assay. >>

- As indicated, we obtained the EM images systematically between pH 8.0 and 4.0 at intervals of 0.5 pH units. This gallery of images is included as Fig. 6b in the revised manuscript, and is positioned next to the turbidity assay results. Considering the contribution of the EM specialists, we included them as co-authors in the revised manuscript.

<< Minor points

3. It would be interesting to see negative stain TEM of the R-BiP:p62 complex at some selected pH values. >>

- We performed negative stained TEM experiments on the R-BiP:p62 complex at pH 8.0 and 6.0. Since R-BiP comprises a 46 kDa monomer in solution (based on our SEC-MALS results), it is difficult to visualize. The EM image of the mixture is almost identical to that of p62 alone at both pH 8.0 and 6.0. We were interested in whether this complex forms large clusters as demonstrated between p62 and ubiquitylated cargos (Zaffagnini *et al.* 2018), but it does not. The UBA domain and ubiquitylated cargo interaction must be very critical for this process, which we noted shortly in the revised manuscript (- line 396, page 18, “According to a recent paper....crosstalk with autophagy machinery”; line 409, page 19, “The interaction between the ubiquitylated cargos....form a cluster with greater efficiency.)

<< 4. Are the authors able to rationalize the pH dependence of p62 polymerization by an analysis of the protonation states of critical residues involved in the electrostatic interactions crucial for PBI-PBI interactions? Such residues are K7, R21, D69 and E82. >>

- Yes, the charged residues indicated by reviewer #2 were all critical for oligomerization. We generated R21A and E82A mutants and performed gel filtration experiments. As shown in the figure below (next page), they behaved as monomers (or mixture with dimers) in solution. In contrast to other mutants, R21A mutant exists exclusively as a monomer.

<< 5. Comparing Fig 2a and Fig 5a it is clear that in Fig 5a the authors should include FP binding studies with Flag-ZZ at pH 6.0 to control that the pH effect on binding to GST-ZZ is mediated by the actual binding between the degron peptide and the ZZ domain and not an indirect effect mediated by pH effects on GST. >>

- As suggested by the reviewer, the FP experimental data on the Flag-ZZ construct at pH 6.0 have been included as Fig. 5a in the revised manuscript.

<< 6. The term oligomer is used throughout for polymers of p62. When the MW is around 1 MDa (lines 314-315) it is more relevant to use the term polymer rather than oligomer as there must be more than ten p62 molecules forming a 1 MDa polymer. >>

- As indicated by the reviewer, the term oligomer for large-sized p62 with MW of 1 MDa is awkward and thus, it has been replaced with the term polymer in many parts of the revised manuscript.

<< 7. Ref 8 should be added in line 78 following “...is recognized by the ZZ domain of p62;” >>

- **Ref. 8 has been included in the position referred to above.**

<< 8. Line 161: delete “approximately” >>

- **This has been deleted.**

9. Line 186: Ref 38 should be added along with ref 40

- **Ref. 38 has been included in the position referred to above.**

<< 10. Line 283: “higher” should be corrected to “lower”>>

- **This has been corrected to “lower”. Thank you.**

Reviewers' comments:

Reviewer #1 (Remarks to the Author):

The authors performed lots of additional experiments and addressed many of the concerns. However, there still remain concerns about in vivo data. The authors did not respond to the comments "In cells without HA-p62, degradation of R*-BiP was observed. Is this degradation mediated by endogenous p62? If so, why expression of HA-p62 mutants impaired the degradation of R*-BiP? Did they function as dominant-negative?". Moreover, the authors did not respond to the comment "Perform control experiment under autophagy-inhibited conditions" and did not provide any data revealing that the observed degradation of R*-BiP and p62 are dependent on autophagy. In order for that, study the effect of autophagy inhibitors on the level of R*-BiP and p62. Minor point: why did authors draw R*-BiP as oligomers in the right (acidic pH) of Figure 7b? Are there any experimental evidence for that?

Reviewer #2 (Remarks to the Author):

The authors have addressed my questions/criticisms very satisfactorily and revised their paper accordingly. I find the revised version to be acceptable for publication.

Responses to Referee #1

<<< *The authors performed lots of additional experiments and addressed many of the concerns. However, there still remain concerns about in vivo data. The authors did not respond to the comments “In cells without HA-p62, degradation of R*-BiP was observed. Is this degradation mediated by endogenous p62? If so, why expression of HA-p62 mutants impaired the degradation of R*-BiP? Did they function as dominant-negative?”. Moreover, the authors did not respond to the comment “Perform control experiment under autophagy-inhibited conditions” and did not provide any data revealing that the observed degradation of R*-BiP and p62 are dependent on autophagy. In order for that, study the effect of autophagy inhibitors on the level of R*-BiP and p62. >>>*

- We regret that we missed some points raised by reviewer. Yes, we believe the R*-BiP is degraded by endogenous p62 in the cells without HA-p62. The R*-BiP was markedly degraded by the over-expressed HA-p62, but not by the mutants (oligomerization or binding defect). Therefore, it is a dominant-negative effect, which the over-expressed mutants interfere the degradation function of tiny amount of endogenous p62. Furthermore, as reviewer suggested, we have performed the same experiments with or without autophagy inhibitor, chloroquine. As shown in Supplementary Fig. 7 in the 2nd revised manuscript (also shown below), the degradation of HA-p62 WT with R*-BiP substrate can be clearly observed in normal condition. However, the R*-BiP substrate as well as p62 were not degraded in presence of chloroquine. Therefore, the degradation of R*-BiP and p62 clearly depends on autophagy.

<<< **Minor point:** why did authors draw R*-BiP as oligomers in the right (acidic pH) of Figure 7b? Are there any experimental evidence for that? >>>

- As shown in Figure 6c (turbidity assay), R*-BiP protein became aggregates at acidic pH. Therefore, we drew R*-BiP as oligomers.

REVIEWERS' COMMENTS:

Reviewer #1 (Remarks to the Author):

The authors have addressed all my concerns.